# Amino acids bind to phase-separating proteins and modulate biomolecular condensate stability and dynamics

Xufeng Xu[1] ✉, Merlijn H. I. van Haren[1], Iris B. A. Smokers[1], Brent S. Visser[1], Paul B. White[1], Robert S. Jansen[2] & Evan Spruijt[1] ✉

Biomolecular condensates (BCs) are ubiquitous compartments that regulate key functions in cells. BCs are surrounded by a complex intracellular environment, of which amino acids (AAs) are prominent components. However, it is unclear how AAs interact with condensate components and influence the material properties of condensates. Here, we demonstrate that phase separation is suppressed with glycine by using model heterotypic condensates composed of nucleophosmin 1 and ribosomal ribonucleic acid. The condensate density decreases, and the dynamics within the condensate increase. We find that glycine weakly binds to amide groups in the protein backbone and aromatic groups in the side chains, weakening the backbone-backbone interactions between neutral and charged disordered proteins while strengthening the interactions between aromatic stickers. This leads to different modulations of the phase behaviour in condensates formed by π/cation-π interactions and charge complexation. We further show that a modulation effect on BCs is observed for other proteinogenic AAs and can be transferred to short homopeptides. These insights offer strategies to modulate the dynamic properties of BCs in vivo.

Biomolecular condensates (BCs) are condensed bodies in the cell, which normally form through liquid-liquid phase separation[1]. They are composed of a rich variety of proteins and nucleic acids[2] and are found in a wide range of cell types and developmental stages[3]. The largest and earliest observed BC is the nucleolus, which was reported in the 1830s[4]. Only recently, the formation process and physical properties of these BCs were studied systematically by Brangwynne, Hyman, Rosen, and others[5–7]. Subsequently, the fundamental roles of BCs in cellular homoeostasis and disease have been extensively studied[8,9]. The phase-separated nature of BCs makes them sensitive to various types of changes in their environment, including crowding, pH, ionic strength, and the presence of modulators in the form of ATP, peptides, and other guest molecules[10–12].

Amino acids (AAs) are known to constitute a major component of the intracellular milieu. Strikingly, more than 25% of the total volume

and 6% of the total dry mass of a mammalian cell was reported to be taken up by free AAs[13]. It has been recently found that AAs have a general effect on modulating protein-protein interactions[14,15], as demonstrated using a range of analytical techniques[16,17], especially analytical ultracentrifugation[18–21]. Some AAs are also found to modulate the formation process of stress granules in vivo[22], a type of cytoplasmic condensates that form in response to cellular stress. It has also been shown that specific AAs can impact the formation of BCs in vitro. Paccione et al.[23] reported that glutamate (E) enhanced the condensate formation of bacterial cell division protein FtsZ and its DNA-bound regulator SlmA. However, it remains unclear how AAs interact with components that form BCs and how the material properties of BCs are modulated as a result of these interactions. Given that AAs have been reported to modulate various biomolecular condensate systems, we hypothesize that free AAs may interact with condensate

[1]Institute for Molecules and Materials, Radboud University, Nijmegen, The Netherlands. [2]Department of Microbiology, Radboud Institute for Biological and Environmental Sciences, Radboud University, Nijmegen, The Netherlands. ✉e-mail: xfengxu@hotmail.com; evan.spruijt@ru.nl

components, partition into condensates, thereby influencing their stability and dynamic properties.

In this work, we study the effect of glycine on the phase separation of nucleophosmin (NPM1) and ribosomal ribonucleic acid (RNA), which is an in vitro heterotypic condensate model of nucleoli[24]. Indeed, AAs are abundant within the nucleus[25] and have been implicated in the regulation of nucleolar function and nuclear DNA replication[26]. For NPM1-RNA in vitro condensates, we find that the miscibility gap decreases, while protein dynamics inside the condensates increase with the addition of glycine. In addition to glycine, we also tested other proteinogenic AAs. Interestingly, except for glutamate (E), all tested amino acids have a dissolution effect on NPM1-RNA condensates, suggesting that an interaction between the AA backbone and the condensate components underlies the general modulation effect.

NPM1-RNA condensates are formed through a combination of complex interactions[27]. To disentangle the role of specific interaction types in condensate behaviour, we examined four additional model systems driven by singular dominant driving forces: $K_{72}$-ATP[28], polyLys-polyAsp ($K_{10}$-$D_{10}$)[29] (both formed by electrostatic interaction), FFssFF[30] (formed by π-π stacking), and WGR-4 peptide[31] (formed mainly by cation-π interactions). By the combinative use of nuclear magnetic resonance (NMR), liquid chromatography-mass spectrometry (LC-MS), and microplate reader assays, we find that AAs bind weakly to amide groups in the protein/peptide backbone and aromatic groups[32] in the side chain, primarily through AAs' amine group in the main chain. This leads to a preferential partitioning of AAs inside BCs. We also find that short homopeptides with up to 8 amino acids exhibit similar effects on NPM1-RNA condensates at the same AA residue concentration, suggesting that they can bind to protein backbones in an additive manner[14]. These findings open up a molecular design platform to fine-tune the material properties of BCs, which may find applications in regulating protein functions in vivo and treating condensate-related diseases[8,33].

## Results and discussion

### Glycine modulates the stability and material properties of NPM1-RNA condensates

We use a heterotypic condensate model of the granular component of the nucleolus, consisting of NPM1 and ribosomal RNA[34]. NPM1 and RNA undergo liquid-liquid phase separation under physiological pH (10 mM Tris, 150 mM NaCl, pH 7.5), leading to the formation of well-defined spherical condensates, enriched in both NPM1 and RNA (Fig. 1a) with a substantial NPM1 miscibility gap (width of the two-phase region), which agrees with previous findings[24,35]. To investigate if AAs impact NPM1-RNA phase separation, we selected glycine (G) as the simplest AA to minimize additional effects from AA side chains. With increasing concentrations of glycine, NPM1-RNA condensates became gradually less bright and less spherical (Circularity shown in Supplementary Fig. 1a), suggesting that their local density and surface tension decreased (Fig. 1b). We quantified the concentrations of NPM1 in the dilute and condensate phase as a function of glycine concentration using fluorescence spectroscopy and microscopy (experimental details in Methods) and found that the NPM1 concentration in the dilute phase increased by ~50% from 5.9 μM up to 8.6 μM, while the concentration in the condensate decreased 6-fold from 228 μM down to 36 μM, reaching a plateau at around 0.6 M glycine (Figs. 1c, d). However, the partitioning of RNA was affected only slightly (Supplementary Fig. 1b, c), suggesting that glycine weakened NPM1-NPM1 or NPM1-RNA interactions, while RNA-RNA interactions remained unchanged. We also observed a similar effect of glycine on NPM1 condensates formed in the presence of the crowding agent PEG (10 kDa)[35] (Supplementary Fig. 2).

To further quantify the effect of glycine on intermolecular interactions underlying phase separation, we calculated the tie-line gradient k, according to Qian et al.[36], which can be expressed by effective interaction difference for the condensate formation ($\chi^\Delta$): $k \approx -\frac{1}{(1+2\chi^\Delta)N_1\phi_1}$ where $\phi_1$ and $N_1$ denote the volume fraction and length of component 1 (NPM1), respectively. $\chi^\Delta = \frac{z\Delta\mu}{k_BT}$, where $\Delta\mu$ denotes the contact energy difference between solvent-solute (water-NPM1, water-RNA, water-AA) pairs and the average of solvent-solvent and solute-solute interactions, z denotes the coordination constant, $k_B$ denotes the Boltzmann constant, and T denotes the absolute temperature[36]. As shown in Fig. 1e, the tie-line gradient k increases with increasing glycine concentration, indicating a weaker net interaction driving the condensate formation.

The weaker associative interaction in the condensate phase (i.e., decreased condensate stability) in turn modulates the local dynamic properties of the condensates. As shown in Fig. 1f, NPM1 recovers faster after photobleaching at high glycine concentration. The calculated recovery half-life ($t_{1/2}$) for NPM1 in the condensates decreased by more than a factor of two (from 12 s to 5 s) after the addition of 0.9 M glycine. By contrast, $t_{1/2}$ of RNA hardly changed with the addition of glycine (Supplementary Fig. 3). In agreement with the FRAP experiment (Fig. 1g), the effective viscosity of the condensate, as estimated from the diffusion of free fluorescein (Alexa Fluor 488 or A488) molecules (diffusion coefficient data shown in Supplementary Fig. 3b) by raster image correlation spectroscopy (RICS), decreased by a factor of two (from 10 to 5 mPa·s). We also tested whether this altered condensate stability affects the partitioning of client molecules ($(RRASL)_3$/RP3), an arginine-rich peptide that interacts with the condensates via electrostatic force[37]. Fluorescein-labelled RP3 exhibited a significant drop in partition coefficient ($K_p$) from 30 to 8 after the addition of 0.9 M glycine (Fig. 1h, representative fluorescence images for RP3 in the NPM1-RNA condensates shown in Supplementary Fig. 1d).

### Glycine dissolves synthetic condensates driven by electrostatic interaction and promotes those driven by π-π stacking and cation-π interaction

To deconvolute the interplay of glycine with the complex multimodal interactions[38] that drive the formation of NPM1-RNA condensates, four model synthetic condensates with simplified driving forces were employed. As shown in Fig. 2a, the lysine(K)-rich elastin-like protein GFP-GFPGAGP[GVGVP(GKGVP)$_9$]$_8$GWPH$_6$ (K72 in short), which contains 72 repeats of the pentapeptide GKGVP (an elastin-like sequence)[39] fused to an N-terminal green fluorescent protein (GFP) for visualization purposes, can form condensates at a low concentration (10 μM) with ATP by electrostatic interaction under physiological pH (25 mM HEPES, pH 7.4)[28]. We measured the K72 concentration in the dilute phase by the fluorescence emission from the conjugated GFP, and we found that with higher glycine concentrations in solution[40], K72 concentrations in the dilute phase increased (Fig. 2a). This agreed with a lower K72 partitioning in the condensate phase (Supplementary Fig. 4), which indicates a gradual condensate dissolution process with the addition of glycine. By measuring protein concentration changes in the dilute phase, we also found that the effect of glycine on the phase behaviour of K72-ATP is qualitatively similar to that of NPM1-RNA (Supplementary Fig. 5), although they are dominated by different condensate-driving forces.

A second synthetic model condensate that is driven by electrostatic interaction consists of two short peptides, oligo-lysine ($K_{10}$) and oligo-aspartic acid ($D_{10}$)[29]. We measured the $K_{10}$ concentration in the dilute phase by the fluorescence emission from the fluorescein-labelled $K_{10}$ and found again that with higher glycine concentration in solution, $K_{10}$ concentration in the dilute phase increased, which also indicates a similar condensate dissolution process (Fig. 2b). This also agreed with a lower $K_{10}$ partitioning in the condensate phase (Supplementary Fig. 6).

We then switched to model condensates formed by π-π stacking and cation-π interaction, which are also among the major driving forces for condensate formation[6,41]. We chose a minimal sticker and

spacer architecture model in the form of FFssFF, which contains two diphenylalanine stickers[30]. FFssFF molecules are soluble in acidic pH due to their net positive charge and start to form condensates due to π-π stacking when the pH is increased above approximately 6.5. We found that with increasing glycine concentration (buffer pH change after the addition of glycine summarized in Supplementary Table 1), the concentration of FFssFF in the dilute phase decreased, and condensates formed at a lower pH, compared to the control without

glycine addition (Supplementary Fig. 7). Accordingly, the turbidity of the whole phase-separating solution increased at a lower pH (Supplementary Fig. 7), which indicates a promotion of condensate formation with the presence of glycine.

Another minimalistic homotypic peptide condensate based on π-π and cation-π interactions was employed. The decapeptide W(GR)₃GWY (WGR-4) was reported by Lampel and co-workers[31] to form condensates at neutral pH due to cation-π attractions between

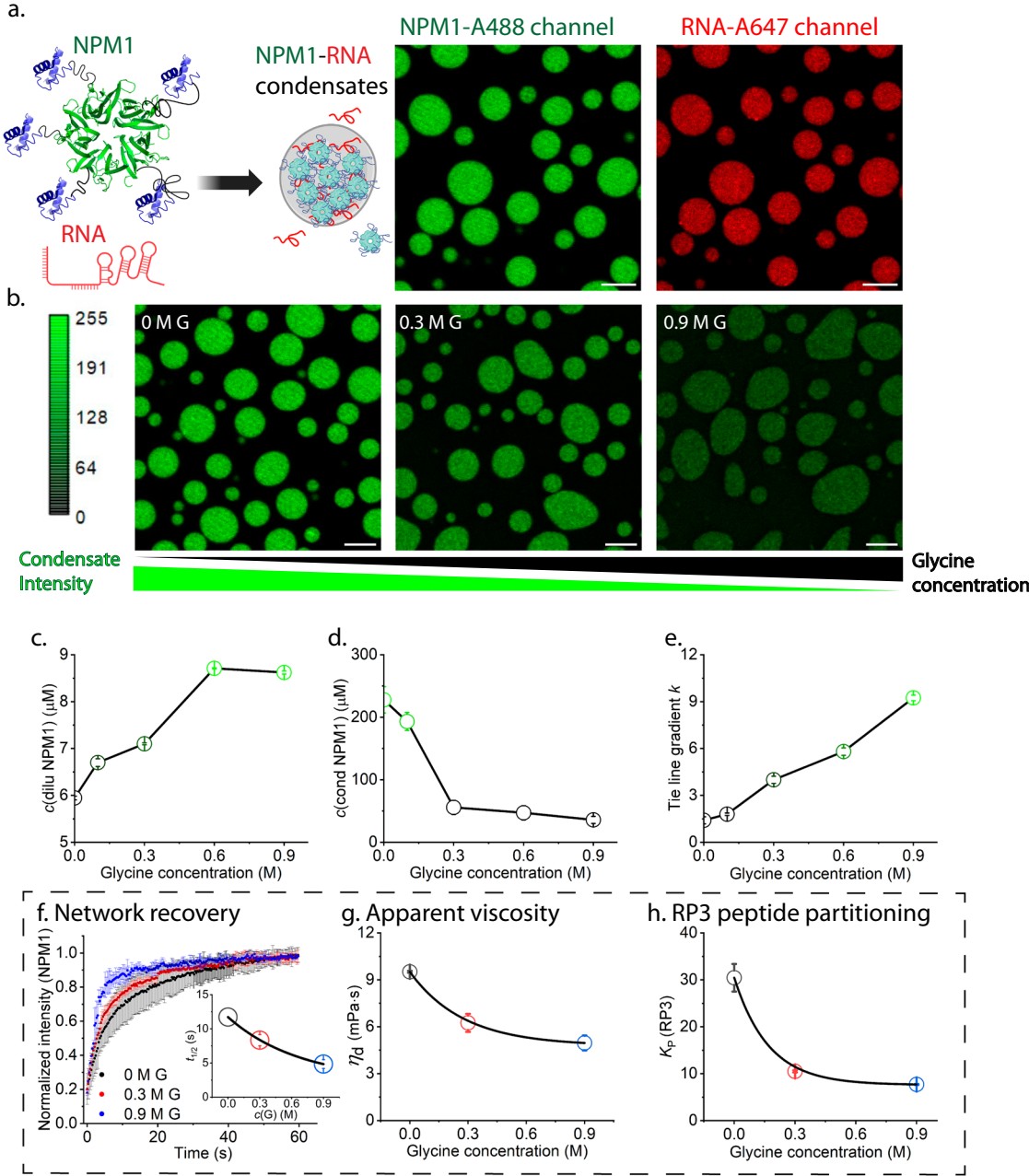

**Fig. 1 | The phase behaviour and material properties of NPM1-RNA condensate after the addition of glycine. a** Schematic illustration of NPM1 protein structures (oligomerization domain (green, PDB: 4N8M) connected via disordered regions (grey) to the C-terminal nucleic acid binding domain (blue, PDB: 2VXD) and their formation of condensates with RNA. Fluorescence confocal microscopy images of NPM1-RNA condensates in NPM1-A488 channel and RNA-A647 channel. Cartoons were created in BioRender (Stellacci, F. (2025) https://BioRender.com/n51cawm). **b** Confocal fluorescence microscopy images of NPM1-RNA condensates in NPM1-A488 channel after the addition of 0, 0.3, and 0.9 M glycine (laser power: 50%, λ(excitation): 485 nm at 25% intensity, the colour bar on the left); **c, d** NPM1

concentrations in the dilute (abbreviated as "dilu") and condensate (abbreviated as "cond") phases after the addition of glycine (0, 0.1, 0.3, 0.6, and 0.9 M) (calculation details in Methods); **e** Calculated tie line gradient k after the addition of glycine (0, 0.1, 0.3, 0.6, and 0.9 M); **f** Average FRAP recovery curves of NPM1 and the calculated recovery half-life (t₁/₂) after the addition of glycine (0, 0.3, and 0.9 M); **g** Apparent viscosity (η_d) of fluorescein (Alexa Fluor 488 or A488) in NPM1-RNA condensates after the addition of glycine (0, 0.3, and 0.9 M); **h** Partition coefficients (K_p) of RP3 in NPM1-RNA condensates after the addition of glycine (0, 0.3, and 0.9 M). Data are expressed as mean ± standard deviation of n = 3 independent experiments. Scale bar = 5 µm for all the fluorescence confocal microscopy images.

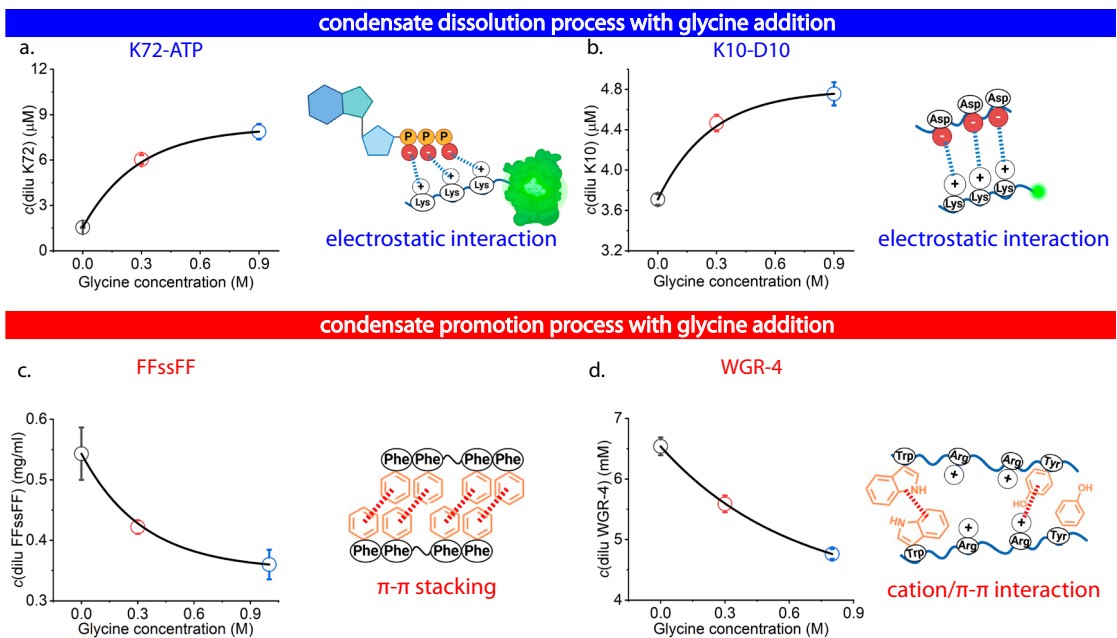

**Fig. 2 | Model synthetic condensates to deconvolute the complex interaction in NPM1-RNA condensates.** The peptide/protein concentration in the dilute phase after the addition of glycine (0, 0.3, and 0.9 M) as well as the expected intermolecular interactions underlying the condensate formation of **a.** K72-ATP system; **b** K10-D10 system; **c** FFssFF system at pH 6.8 (the FFssFF concentration in the dilute phase and the turbidity of the whole solution at pH from 6 to 8 shown in Supplementary Fig. 7) and **d** WGR-4 system. Data are expressed as mean ± standard deviation of $n = 3$ independent experiments. Cartoons were created in Biorender (Stellacci, F. (2025) https://BioRender.com/p727lpy).

arginine residues (R) and aromatic residues (W and Y) as well as π-π stacking among aromatic residues (W and Y). Similar to the FFssFF system, we also found that with increasing glycine concentration, the WGR-4 concentration in the dilute phase decreased (Fig. 2d) and the turbidity of the solution increased (Supplementary Fig. 8), which both indicate a promotion of condensate formation with the addition of glycine, similar to the FFssFF system.

**Backbone and aryl binding of amino acids underlie the modulation of condensate stability**

The curves of increasing dilute phase NPM1 concentration with increasing glycine concentration are reminiscent of a binding isotherm[15,42]. In order to estimate the order of magnitude of the effective affinity between the AA and the protein, we used a Langmuir-type binding model[42] to fit the NPM1 protein concentration change in the dilute phase $\Delta c$(dilu NPM1) as a function of glycine concentration. We found an apparent dissociation constant ($K_d$) of glycine of ~1.0 M (Fig. 3a), corresponding to weak binding affinity. This weak apparent binding between glycine and the condensates agrees with the recently proposed theory that AAs can weakly bind to protein patchy surfaces and modulate protein-protein interactions by effectively screening a fraction of their attractive interaction potential[15]. We also measured the binding affinity of four different AAs (glycine/G, proline/P, serine/S, and alanine/A) to K72-ATP condensates (Supplementary Fig. 9). All the fitted values of $K_d$ were found to be ~1 M (Fig. 3b).

Weak binding of AAs to NPM1, K72, and possibly other phase-separating proteins suggests that AAs may accumulate inside the condensates. We therefore employed LC−MS and ¹H NMR to measure $K_p$ of AAs[43–45]. We found that $K_p$ for all the tested AAs are ~4 in NPM1-RNA condensates and ~6 in K72-ATP condensates (Fig. 3c, d, NMR spectrum with peak assignments in Supplementary Fig. 10 and LC-MS raw data in Supplementary Fig. 11), in reasonable agreement with an apparent dissociation constant of ~1 M and an estimated local protein concentration of 1 mM for K72[28].

The Langmuir-type binding model suggests that AAs bind to specific sites along the protein. To elucidate the binding positions of

AAs on BCs, we employed NMR spectroscopy under conditions where phase separation does not occur, as was previously reported for ion binding to condensate-forming biomolecules[46]. Following the assignments of proton peaks by the Total Correlation Spectroscopy (TOCSY) (Supplementary Fig. 12), we ran ¹H NMR experiments by the titration of deuterium-labelled glycine (G-d⁵) into solutions of K72-GFP. We observed significant changes in backbone amide chemical shifts of glycine/G and valine/V residues in K72, while the chemical shifts for the other proton peaks hardly changed, which indicates the proximity or binding of G to backbone amide groups (Fig. 4a). A linear chemical shift perturbation (CSP) indicates a change in the local environment, possibly due to changes in solvation, while a nonlinear CSP is an indication of specific binding. To estimate an apparent binding constant from the observed nonlinear CSP, the chemical shift perturbation (CSP) at different concentrations of G-d⁵ was also fitted with the Langmuir-type binding model[42] and $K_d$ of G-d⁵ and amide groups in G and V residues was estimated to be $1.5 \pm 0.2$ and $1.7 \pm 0.3$ M, respectively (Fig. 4d). The overall $K_d$ of G-d⁵ to K72 can be expressed as: $K_{d(overall)} = \frac{c(G)c(K72)}{c(K72-\text{bound to G})+c(K72-\text{bound to V})}$, which leads to the relation: $\frac{1}{K_{d(overall)}} = \frac{1}{K_{d1}} + \frac{1}{K_{d2}}$. Therefore, the overall $K_d$ can be estimated to be ~0.8 M, which agrees well with the binding affinity obtained from $\Delta c$(K72) in the dilute phase ($0.6 \pm 0.2$ M) (Fig. 3b).

Interestingly, the ¹H NMR experiments of FFssFF and WGR-4 after the addition of G-d⁵ also show significant changes in backbone amide group chemical shifts. Moreover, we also observed significant and apparently nonlinear changes in side-chain aromatic group chemical shifts (Fig. 4b, c). The chemical shifts for the other groups remained unchanged (TOCSY in Supplementary Fig. 13 for the amide and aromatic group assignments). The binding affinity of glycine to aromatic groups is also similar to that of amide groups, both characterized by an apparent $K_d$ of ~1.5 M (Fig. 4e). Overall, the chemical shift data of these peptides show a consistent perturbation of backbone amide groups by glycine, which impacts intermolecular amide-amide hydrogen bonds. The additional perturbation on side chain aromatic groups is observed for peptides rich in aromatic AAs, which has additional effects on inter-

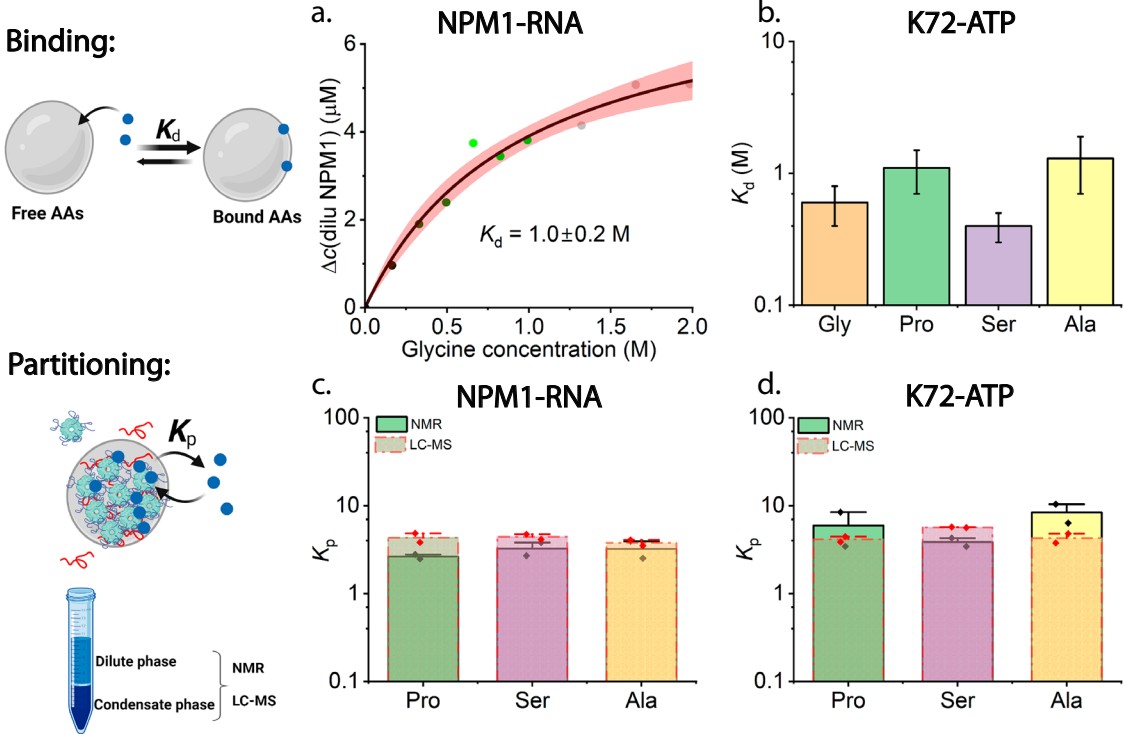

**Fig. 3 | Binding and partitioning of AAs in NPM1-RNA and K72-ATP condensates. a** The NPM1 protein concentration in the dilute phase after adding glycine at varying concentrations and the fitting curve (solid black) with 95% confidence band (red) using the Langmuir-type binding model; **b** The fitted binding affinities ($K_d$) for four different AAs (glycine, proline, serine, and alanine) on K72-ATP condensates ($K_d = 0.6 \pm 0.2$ M for glycine, $1.1 \pm 0.4$ M for proline, $0.4 \pm 0.1$ M for serine, and $1.3 \pm 0.6$ M for alanine); The measured partition coefficients ($K_p$) by both NMR and LC-MS for three representative AAs (proline, serine, and alanine) in the condensate phase of **c.** NPM1-RNA systems and **d.** K72-ATP systems. Data in a and b are from $n = 1$ independent experiment. Data in b are expressed as mean ± standard deviation of the fitting result, shown in Supplementary Fig. 9. Data in c and d are expressed as mean ± standard deviation of $n = 2$ independent experiments. Cartoons were created in Biorender (Stellacci, F. (2025) https://BioRender.com/sp9nze0).

molecular π-π stacking and cation-π interactions. Thus, we hypothesize that the binding of AAs increases the effective dielectric permittivity in the vicinity/microenvironment around the binding sites[47–49]. This leads to the weakening of protein backbone-backbone hydrogen bondings, which have so far not received much attention in the context of condensate formation, and the strengthening of π-π and cation/π interactions. Overall, AAs can thus enhance condensation driven predominantly by cation/π-π interactions, and suppress condensation driven by charge complexation.

## Condensate modulation extends to most proteogenic amino acids and is transferable to short homopeptides

To investigate whether the modulation effect of glycine is a general property of AAs, we screened all the neutral proteogenic AAs with a solubility > 100 mM as well as charged AAs for their effect on NPM1-RNA, K72-ATP, and WGR-4 condensate systems using a high-throughput microplate reader assay. As shown in Fig. 5a, all the AAs tested show a condensate dissolution effect on NPM1-RNA, as indicated by higher protein concentrations in the dilute phase after the addition of AAs. The only exception is glutamic acid/glutamate (E). We found slightly lower (~2%) protein concentrations in the dilute phase after the addition of 20 mM E compared to the control set, indicating enhanced condensate formation. Similarly, except for E, all the other AAs tested show a condensate dissolution effect on K72-ATP system. The enhanced condensate formation by the addition of E is likely due to its preferential exclusion from peptide backbones and side chains, thereby promoting protein–protein association and stabilizing the condensate phase[50,51]. E may be exploited by cells to promote phase separation of specific proteins, potentially as a mechanism to regulate condensate formation in vivo[23,50,51]. A

condensate promotion effect was also observed for all the tested proteinogenic AAs on WGR-4 systems, except for one outlier, proline (Supplementary Fig. 14).

Furthermore, two short homopeptides, a glycine trimer $(G)_3$ and proline trimer $(P)_3$, were employed to investigate if they could also modulate condensate formation, as both glycine and proline are among the most water-soluble AAs. As shown in Fig. 5c–f, very similar modulation effects on the NPM1-RNA phase separation were observed for these two short homopeptides at the same AA residue concentrations as free AAs, which indicates that the modulation effect may be transferable to homopeptides in an additive manner[14]. Moreover, the linear transferability in the modulation effect is still true for proline octamer $(P)_8$, whereas data for the glycine octamer $(G)_8$ could not be obtained due to its very low solubility (Supplementary Fig. 15a, b). However, aggregation started to appear (Supplementary Fig. 15c, d), which may be due to the molecular rigidity of a relatively long peptide chain compared to free AAs.

## Towards a molecular understanding of amino acid-mediated condensate modulation

Based on the experimental data presented in this study, we propose a multiscale mechanism behind the macroscopic modulation effect of AAs on BCs, Fig. 6. On a molecular scale, AAs bind to backbone amide groups of phase-separating proteins, which weaken the intermolecular amide-amide hydrogen bonds. This explains that AAs can dissolve NPM1-RNA, K72-ATP, and $K_{10}$-$D_{10}$ condensates. We also find that most AAs have modulation effects (Fig. 5). This suggests that the binding is mediated primarily through the free amine or carboxylic acid group in the AAs' main chain. To shed more light on the binding mechanism, we

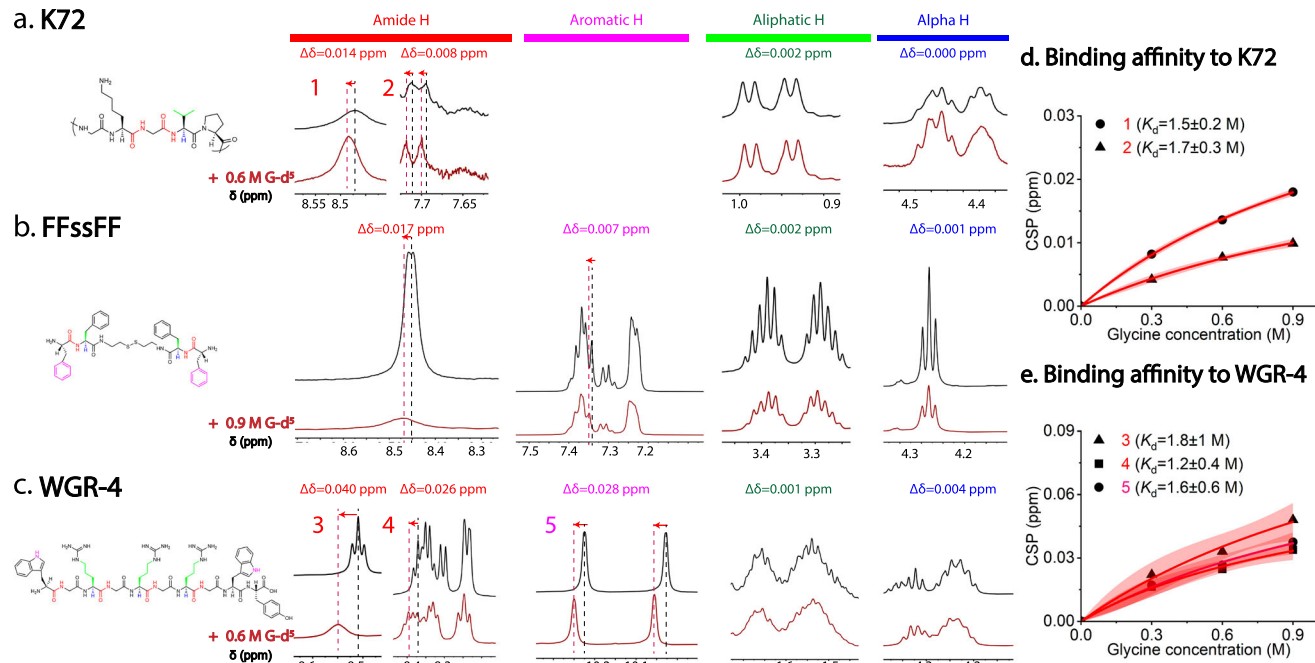

**Fig. 4 | Binding of glycine (G-d⁵) to different proteins/peptides from ¹H NMR spectroscopy.** The protein/peptide chemical structures, the proton peaks with significant chemical shifts (in red and pink) and without significant chemical shift perturbations (in green and blue) after the addition of G-d⁵ for **a.** K72; **b** FFssFF and **c.** WGR-4; The chemical shift perturbation (CSP) of **d.** K72 and **e.** WGR-4 with the titration of G-d⁵ at 4 different concentrations to estimate the binding affinity ($K_d$) of G to K72 and WGR-4. Data in d and e are from $n = 1$ independent experiment, and the fitting curves are shown in solid red with 95% confidence band (red) using the Langmuir-type binding model.

investigated two glycine derivatives: betaine, which contains a tertiary amine instead of a primary amine, and taurine, which contains a sulfonic acid instead of a carboxylic acid group. We found that adding betaine did not affect NPM1-RNA condensate formation, while adding taurine similarly dissolved condensates as glycine (Supplementary Fig. 16). This indicates that the primary amine group of AAs most likely binds to the proteins in BC components through hydrogen bonding.

In addition, we employed synthetic condensates formed by the electrostatic interaction between poly(diallyldimethyl-ammonium chloride) (PDDA) and poly(acrylic acid) (PAA), which do not contain amide groups in their backbones. We found that the addition of glycine did not affect the phase separation of this system (Supplementary Fig. 17a). We also found that there was no preferential partitioning of AAs into the condensate phase (Supplementary Fig. 17b). This experiment further supports our hypothesis that AAs bind to amide groups in the backbone of disordered proteins and modulate the protein backbone-backbone interactions, which contribute to condensate stability. AAs do not suppress phase separation when there are no amide groups in their backbone to bind to, such as in the case of PDDA-PAA condensates.

According to our proposed molecular mechanism, no specific AA side chains are needed for the observed interaction with BC components. This is supported by our findings that most AAs have modulation effects (Fig. 5). More importantly, the observed transferability of the modulation to short homopeptides indicates that peptides can bind reasonably strongly to disordered protein backbones, and opens the way for the rational peptide design to modulate BCs. This may be realized by strategically engineering side-chain interactions at specific binding sites. We can thus harness this biocompatible and versatile peptide platform to specifically target the properties and functions of physiologically important biomolecular condensates in the future[52–54]. These strategies may help to control the biological processes related to BCs, and potentially

suppress undesired aging of condensates, which has been linked to a variety of neurodegenerative diseases[8,55–58].

## Methods
### Materials
All chemicals and reagents were used as received from commercial suppliers. The following chemicals were purchased from Sigma Aldrich: adenosine triphosphate (ATP), sodium chloride, Tris base, PEG (10k Da), and all the 18 amino acids that are used in the study, including G, S, T, Q, C, P, A, V, M, H, N, I, L, F, R, K, D and E and glycine-d₅(175838). poly(acrylic acid) sodium salt (PAA, 15 kDa, 35 wt% solution in $H_2O$) and poly(diallyldimethylammonium chloride) (PDDA, 200–350 kDa, 20 wt % solution in $H_2O$). HEPES-free acid was purchased from FluoroChem. PLL-$g$[3.5]-PEG was purchased from SuSoS. Poly-L-lysine hydrobromide (MW = 2100 Da, 10-mer), fluorescein isothiocyanate (FITC)-labelled poly-L-lysine (10-mer), and poly-L-aspartic acid sodium salt (MW = 1400 Da, 10-mer) were purchased from Alamanda Polymers. WGR-4, (G)₃, (P)₃, and (P)₈ were all purchased from Genscript Biotech (The Netherlands, The Hague). 5,6-FAM-RP3 was purchased from CASLO, Kongens Lyngby, Denmark. GFP-labelled K72 and NPM1 were expressed and purified as previously described[24,28]. FFssFF was synthesized as previously described[30]. *E. coli* ribosomal RNA was purified as previously described[24].

### Condensate preparation
NPM1-RNA condensates were prepared in Tris buffer (final concentration 10 mM, pH 7.5) with 150 mM NaCl, by adding PEG 10 kDa (final concentration 2.3 wt%) and RNA/RNA-A647 (final concentration 100 ng/µL, 1:19 molar ratio Alexa Fluor 647-labelled or totally unlabelled) to Tris buffer followed by NPM1/NPM1-A488 (final concentration 20 µM, 1:19 molar ratio Alexa Fluor 488-labelled).

K72-ATP condensates were prepared in HEPES buffer (final concentration 25 mM, pH 7.4), by adding GFP-labelled K72 (final

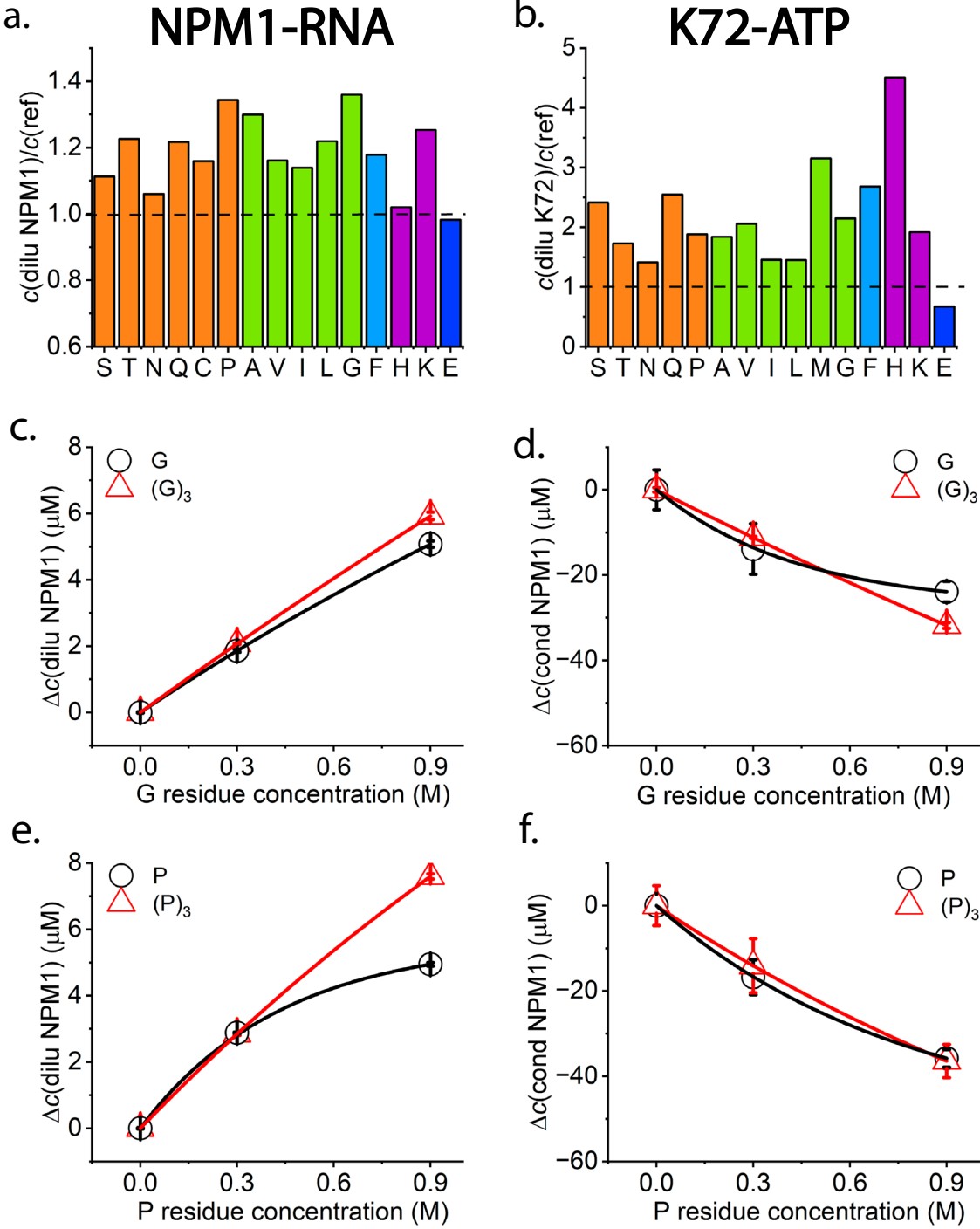

**Fig. 5 | General modulation effects of AAs and the transferability to short homopeptides. a** NPM1 concentration in the dilute phase after the addition of different AAs (200 mM of S, T, Q, C, G, P, A, V; 100 mM of H, N, I, L, F; 20 mM K, E) divided by reference NPM1 concentration in the dilute phase in the absence of any AAs; **b** K72 concentration in the dilute phase after the addition of different AAs (200 mM of S, T, Q, G, P, A, V; 100 mM of M, H, N, I, L, F; 20 mM K, E) divided by reference K72 concentration in the dilute phase in the absence of any AA; AAs are groups in colours by side chain properties. Orange, green, blue, and purple bars represent polar uncharged, nonpolar aliphatic, nonpolar aromatic, and slightly positively charged side groups, respectively; **c, d** NPM1 concentration changes in the dilute and condensate phase after the addition of G and (G)$_3$ at the same ionic strength; **e, f** NPM1 concentration changes in the dilute and condensate phases after the addition of P and (P)$_3$ at the same ionic strength. Data are expressed as mean ± standard deviation of $n$ = 3 independent experiments.

concentration 10 μM) to HEPES buffer, followed by ATP (final concentration 4 mM).

K10-D10 condensates were prepared in HEPES buffer (final concentration 50 mM, pH 7.4), by adding D10 (final concentration 5 mM) to HEPES buffer, followed by fluorescein isothiocyanate (FITC)-labelled K10 (final concentration 5 mM, 1:49 molar ratio of labelled).

10 mg/ml FFssFF stock solution in water was directly diluted 10 times with pH buffers of the pH values from 5 to 8 to form condensates.

10 mM WGR-4 stock solution in water (final concentration 7 mM) was added to the Tris buffer (final concentration 20 mM, pH 7.5, 800 mM NaCl) to form condensates.

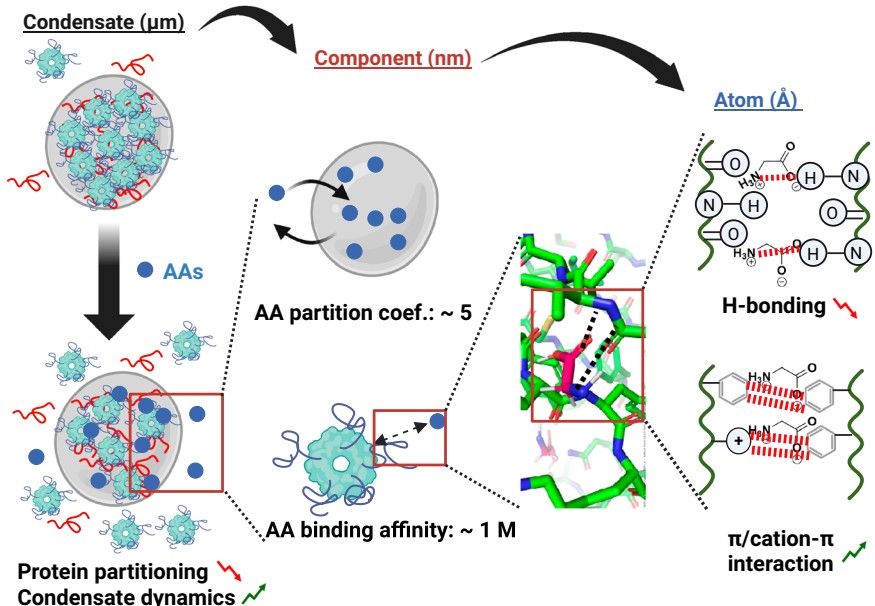

**Fig. 6 | Multiscale modulation.** The proposed multiscale mechanism for the modulation effect of AAs on BCs, ranging from μm-scale condensate formation and dynamics, nm-scale component interaction and partitioning to Å-scale atomic interaction. Created in Biorender (Stellacci, F. (2025) https://BioRender.com/qaa1ziw).

## Confocal fluorescence microscopy

All samples were imaged on Ibidi 18-well chambered slides (#1.5) that were cleaned with a plasma cleaner, incubated for 24 h with 0.1 mg mL −1 PLL-g[3.5]-PEG (SuSoS, Dübendorf, Switzerland) dissolved in Milli-Q water, and washed and dried with Milli-Q water and pressurized air, respectively. Before image acquisition, the condensate dispersion was transferred to the channel and incubated for over 0.5 h to allow condensates to coalesce and settle on the glass surface. Confocal fluorescence images were acquired on a Leica Sp8x confocal inverted microscope (Leica Microsystems, Germany) equipped with a DMi8 CS motorized stage, a pulsed white light laser, and 2 × HyD SP GaAsP and 2× PMT detectors. Images were recorded using the LAS X v.3.5 acquisition software, using an HC PL APO 100×/1.40 oil immersion objective. The calculation of partition coefficients was performed by the following equation: $K_p = (I_{condensate} - I_{background})/(I_{dilute} - I_{background})$, where $I_{condensate}$ denotes the average intensity of condensates in one image, and $I_{dilute}$ denotes the average intensity of the area without condensates, $I_{background}$ denotes the background intensity by measuring only the buffer at the same settings as for the fluorescent images, which is normally zero.

## Quantification of the protein concentrations in the dilute and condensate phases

A typical sample of 38 μL was prepared in 10 mM Tris (pH 7.5) and 150 mM NaCl with 2.3 wt% of PEG, 20 μM NPM1/NPM1-A488 (1:19 molar ratio labelled), and 100 ng/μL RNA (unlabelled) as described above. After the incubation for 20 min at room temperature, the condensate phase was separated from the dilute phase by centrifugation at 20,000 g for 20 min at room temperature. The supernatant of 20 μL was then transferred to a 384-well plate (Nunc, flat bottom), and the fluorescence intensity was measured on a plate reader (Tecan Spark M10) at 485/535 nm for NPM1-A488.

Concentrations of the dilute phase were calculated based on calibration curves (Supplementary Fig. 18). Calibration curves were performed using a series of known concentrations of NPM1/NPM1-A488 (1:19 molar ratio labelled) prepared in the same buffer and multi-well plates as the experimental samples. The NPM1 concentration in the condensate phase can then be calculated by the following equation: c(NPM1 in condensate phase) = c(NPM1 in dilute phase) × $K_p$. The

calibration curve for calculating the GFP-K72 concentrations in the dilute phase is shown in Supplementary Fig. 19.

## Turbidity measurements

All turbidity measurements were performed using a plate reader (Tecan Spark M10). Absorbance was recorded across the wavelength range of 450 nm to 650 nm, with 600 nm used as the representative wavelength for turbidity. Measurements were taken immediately after transferring the entire dispersion to a 96-well UV-transparent flat-bottom plate (Nunc), right after the condensate formation by mixing all component solutions.

## Fluorescence recovery after photobleaching

Time-lapse videos were recorded at room temperature on a CSU X-1 Yokogawa spinning disk confocal unit connected to an Olympus IX81 inverted microscope, using an ×100 piezo-driven oil immersion objective (NA 1.3) and 488 and 640 nm laser beams. Emission was measured with a 200-ms exposure time using an Andor iXon3 EM-CCD camera. The acquired images have a pixel size of 141 nm and a field of $512 \times 512$ μm². For the laser bleaching, a small region of interest was selected in the middle of a condensate droplet. The 488 or 640 nm laser line was set to 100% laser power using 20 pulses of 200 μs. The recovery was then imaged at reduced laser intensity with a time interval of 300 ms for 200 times. The exponential decay equation, $I_{normalized} = A(1-e^{-bt})+C$ was used to fit the parameters $A$, $b$, and $C$, and the recovery half-life was determined by the equation: $t_{1/2} = ln(2)/b$, according to a 2D-diffusion model with a fixed boundary[59].

## Diffusion coefficients measured by Raster image correlation spectroscopy (RICS)

The RICS was performed on a Leica SP8 confocal microscope equipped with a single-photon detector. Calibration of the focal volume waist $\omega_0$ was performed using the known diffusion coefficient of A488 of 435 μm²/s (T = 22.5 ± 0.5 °C) in water, and $\omega_z$ was set to 3 times the value of $\omega_0$[60]. All measurements were captured at a resolution of $256 \times 256$ pixels with a 20 nm pixel size using a 63x oil objective. Condensates were measured at 10 Hz line speed with 15 frames acquired per data point. Analysis of autocorrelation curves was done using PAM[61], using the 3D RICS diffusion model as described by Digman and Gratton[62].

An example of raw RICS data and autocorrelation curve fitting is shown in Supplementary Figs. 20 and 21. The apparent viscosity ($\eta$) of the condensates was calculated using the Stokes–Einstein relation, based on the measured diffusion coefficient ($D$) of A488 and its hydrodynamic radius ($R$). The equation used is: $\eta = k_B T / 6\pi D R$, where $k_B$ is the Boltzmann constant, $T$ is the temperature, $D$ is the diffusion coefficient, and $R$ is the hydrodynamic radius of A488, taken to be 1.4 nm.

### Langmuir-type binding model fitting of dilute phase NPM1 concentration change and the chemical shift perturbation with different glycine concentrations

The dilute phase NPM1 concentration change and the chemical shift perturbation (both denoted as $\Delta$) with different AA concentrations ($c$) were fitted by a simple binding model[15,42] under the assumption of excess AA: $\Delta = \frac{\Delta_{max} \times c}{K_d + c}$.

### Sample preparation for the partition coefficient measurements of AAs

In a PCV cell counting tubes (capillary graduations only, no cap, Sigma-Adrich), NPM1-RNA condensates were prepared in Tris buffer (final concentration 10 mM, pH 7.5) with 150 mM NaCl, by adding PEG 10k Da (final concentration 2.3 wt%) and RNA (final concentration 100 ng/μL) to Tris buffer followed by NPM1 (final concentration 20 μM) at a total sample volume of 400 μL. After the incubation at RT for 30 min, the tube was centrifuged at 3200 g for 30 min at RT to spin down the condensate phase. After that, ~0.5 μL of condensate phase was obtained at the bottom of the PCV cell counting tubes. The supernatant (dilute phase) was carefully separated from the viscous condensate phase and stored in a separate tube. The condensate phase was redispersed with 1xPBS in an appropriate volume ratio. The dilute phase was diluted with 1xPBS in the same ratio. All the macromolecules were removed by using Amicon® Ultracentrifuge Filter units (10 kDa cut-off) at 4000 g for 30 min. The filtered solution was ready for the partition coefficient measurements by NMR or LC-MS. Similar procedures were used for measuring the AA partitioning in K72-ATP and PDDA-PAA condensates.

### Partition coefficient and chemical shift perturbation experiments by NMR

NMR samples were prepared by dissolving proteins or peptides (10 μM of K72 and 1 mM of WGR-4 proteins) in 500 μl of 1×PBS buffer (pH 7.2) with 10% $D_2O$ (containing 0.05 wt% 3-(trimethylsilyl)propionic-2,2,3,3-$d_4$ acid, sodium salt as the internal standard for chemical shift referencing). Measurements were conducted on a Bruker Avance III 500 MHz NMR Spectrometer equipped with a Prodigy BB cryoprobe at 298.15 K. 1D-$^1$H experiments were performed using the zgesgp water-suppression pulse sequence with 128 scans and a total relaxation and acquisition of 6.3 s. For the chemical shift perturbation experiments, 1D-$^1$H experiments were performed using the zgesgp water-suppression pulse sequence with 128 scans and a total relaxation and acquisition of 6.3 s, and 2D-$^1$H,$^1$H-TOCSY experiments were performed with 60 ms spin-lock, 64 scans per increment, 512 increments with a 6 kHz spectral window in dimension at 298.15 K. All data was processed in MestReNova 14.

### Partition coefficient measurements of AAs by LC-MS

Proline, serine and alanine concentrations were determined using an Agilent 1290 Infinity II LC system coupled to an Agilent Accurate Mass 6546 Quadrupole - Time of Flight (Q-TOF) mass spectrometer, using a previously described method[63]. In brief, 2 μL samples were injected onto a Diamond Hydride Type C column (Cogent) and separated using a 0.4 mL/min gradient of water with 0.2% formic acid (A) in acetonitrile with 0.2% formic acid (B) as follows: 0–2 min: 85% B, 3–5 min: 80% B, 6–7 min: 75% B, 8-9 min: 70% B, 10–11 min: 50% B, 11–14: 20% B, 14–24:

5% B, followed by 10 min re-equilibration at 85% B. Detection was performed in the positive ionization mode and a mass range of m/z 50-1200. Analyte peaks were extracted using a 20 ppm window and integrated manually. The number of biological replicates $n = 2$.

### Labelling of NPM1, RNA and K72

NPM1-A488 and RNA-A647 labelling were performed as previously described by André et al.[35]. In short, NPM1 proteins were labelled with AlexaFluor488 C5 maleimide dye (Thermo Fisher Scientific) according to the manufacturer's protocol, and the 3′ hydroxyl-end of RNA was labelled with AlexaFluor647 hydrazide (Thermo Fisher Scientific) by using a periodate oxidation reaction according to the manufacturer's protocol.

GFP-labelled K72 was expressed and purified following the protocol reported by Nakashima et al.[28]. In short, BL21(DE3) cells were transformed with the pET25-sFil-K72 plasmid and grown in Terrific Broth. The bacterial cultures were grown at 37 °C till reaching an $OD_{600}$ of 1.5–1.8. Protein expression was induced by cooling cultures to 18 °C to proceed overnight. Cells were pelleted by centrifugation and lysed by sonication, and His-tagged K72 was purified from the soluble fraction using Ni-affinity chromatography followed by size exclusion chromatography (SEC). Purified protein was concentrated, snap-frozen in aliquots, and stored at −80 °C.

### PDDA- PAA condensate preparation

PDDA-PAA condensates were prepared by mixing 60 mM (monomer unit, final concentration) poly(diallyldimethylammonium chloride) (PDDA, 200–350 kDa) and 60 mM (monomer unit, final concentration) poly(acrylic acid) sodium salt (PAA, 15 kDa) in 100 mM Tris buffer (pH 7.5), using stock solutions of 1.24 M PDDA and 3.7 M PAA in 100 mM Tris buffer (pH 7.5). After the incubation at RT for 30 min, the condensate dispersion was centrifuged at 21,130 g for 20 min at RT to spin down the condensate phase. The dilute phase was taken from the supernatant for further $^1$H NMR measurements. The PDDA concentration in the dilute phase was calculated from the methyl group proton peak (3.1–3.4 ppm) in the NMR[64] with a reference sample of a known PDDA concentration.

### Reporting summary

Further information on research design is available in the Nature Portfolio Reporting Summary linked to this article.

## Data availability

The source data generated in this study have been deposited in the Radboud Repository [https://data.ru.nl] under accession code [https://doi.org/10.34973/hmv8-d985]. The data are available under CC-BY-4.0 license. Source data are provided with this paper.

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

## Acknowledgements

The authors acknowledge Dr. R.M. de Graaf for running the LC-MS. X.F.X. acknowledges the Swiss National Science Foundation (SNSF) for financial support (P500PN_222304). E.S. acknowledges funding from a Vidi grant from the Netherlands Organization for Scientific Research (NWO).

## Author contributions

X.F.X. and E.S. conceived the project. X.F.X. designed, performed, and analysed all the experiments. M.H.I.v.H. helped with protein purification, labelling, and microscope data analysis. I.B.A.S. developed original NMR methods and gave feedback on the analysis. P.B.W. helped with NMR experiments. B.S.V. helped with performing and analysing the viscosity measurements of RICS. R.S.J. helped with LC-MS experiments. X.F.X. and E.S. wrote the manuscript, with input and revisions from all authors.

## Competing interests

The authors declare no competing interests.
