## [Transparent Peer Review file · Nature Communications]

Amino acids bind to phase-separating proteins and modulate biomolecular condensate stability and dynamics

Corresponding Author: Professor Evan Spruijt

Version 0:

Reviewer comments:

Reviewer #1

(Remarks to the Author)

In the article "Amino acids bind to phase-separating proteins and modulate biomolecular condensate stability and dynamics", Xu *et al.* investigate the modulation of coacervate properties in the presence of soluble proteogenic amino acids. Using a set of known biomolecular liquid-liquid phase separating systems that rely on varying non-covalent interactions such as electrostatics, cation/ π - π , they explore the differential role of amino acids such as glycine in controlling partitioning, network properties and material characteristics such as viscosity. These analyses are reliant on fluorescence from confocal microscopy, LC-MS data, turbidimetry as well as NMR spectroscopy on the studied *in vitro* systems. The role of backbone interactions with amino acids is highlighted, with a general trend observed for soluble amino acids and short repeat peptides that were investigated. The key findings indicate that high concentrations of soluble amino acids like glycine disfavor LLPS in case of heterotypic condensates (NPM1-RNA/K72-ATP/K10-D10) whereas they promote condensate formation for simple coacervates of peptides like FFsFF/WGR-4. Since amino acids and biomolecular condensates co-exist *in vivo* and it is important to understand and manipulate their materials properties, this work is of interest to the community as it augments existing understanding of the regulation of protein-protein interactions by amino acids. However, the authors should address the following concerns prior to this work being considered for publication-

- 1) Results discussed in Figure 1 a-e, indicate that with increasing glycine concentration in case of the NPM1-RNA condensation, NPM1 solubilization in the dilute phase is favored. The authors add that RNA concentrations are not affected significantly (Figure S1). This would imply that in order to achieve effective charge complexation, some NPM1-RNA interactions would be substituted by zwitterionic glycine. This should change the coordination number z discussed in the tie-line gradient analysis on Page 3 and Figure 1 e. Additionally, why is the glycine not factored into this analysis either as a solvent modulator (0), or an additional solute (3) in spite of high concentration? Can the authors explain why they believe that all other variables in this analysis are constant so as to allow them to conclude that the increase in k is exclusively due to weaker associative interactions?
- 2) Alternatively, if the condensates are enriched in RNA when the glycine concentration is increased from 0 to 0.9 M, why is the partitioning of a basic peptide RP3 (Figure 1 h) not enhanced as the negatively charged RNA should favor partitioning of this cationic peptide?
- 3) The electrostatic complexation driven condensates investigated in Figure 2 (K72-ATP and K10-D10) have fluorophores of vastly different molecular weights with the former tagged with GFP which is itself charged. In Figure 1, a 1:19 molar ratio for fluorescently tagged molecules is reported however it is unclear from the methods section on Condensate Formation, what the molar ratio of the labelled:unlabelled molecules used is for Figure 2 systems?
- 4) In Figure 3, using NMR and LC-MS data, the authors show preferential partitioning of amino acids into the complex coacervates of proteins/peptides. Also, in Figure S17 and Discussion (page 10), the authors show that amide groups in backbones are important for interactions which are absent in the PDDA-PAA system. Do amino acids partition into the PDDA-PAA system in the absence of such backbone amide interactions?
- 5) Chemical shift perturbations in Figure 4 are interesting, with strong evidence for glycine-protein/peptide interactions. Could the authors comment on the changes in line shape including broadening upon addition of 0.6 M G-d5?
- 6) The comparison of Glycine with Betaine and Taurine is unclear in terms of the providing evidence for concluding "that the primary amine group of AAs most likely binds to the proteins in BC components through hydrogen bonding" (Page 9). The sulfonic acid group of taurine is a stronger acid, and capable of participating in H-bonding more readily than the carboxylate group of glycine, likely explaining the strong increase in dilute phase concentration of NPM1. It would likely be insightful to

test *N*-acetylated glycine and *C*-amidated glycine to decouple the role of charge and hydrogen bonding from each termini. 7) Figure S14 shows the anomalous trend with proline for WGR-4 partitioning. Could this possibly be due to the higher pKa for the cyclic NH vs all the free NH₂ termini other in amino acids?

Additionally, the manuscript can benefit from a few minor edits –

- 1) On Page 5, the line “In agreement with FRAP...” should end with a reference to Figure 1g.
- 2) Figure S1 b is missing the concentration units on the Y-axis.
- 3) Figure S6 X-axis label has a typo for concentration.
- 4) Figure 6 and Figure S16: Glycine should be represented in zwitterionic form for consistency with betaine.

Reviewer #2

(Remarks to the Author)

Review of Xu et al, Spruit.

The MS titled “Amino acids bind to phase-separating proteins and modulate biomolecular condensate stability and dynamics” focuses on the timely and highly interesting phenomenon of biomolecular assemblies and aims to dissect and analyze the ways in which this behavior is modulated by amino acids. Briefly, the study addresses the following:

- (i) The ‘biological’ NPM1-RNA condensate system is shown to be modulated by high concentrations of glycine. This is established by an array of experiments that are de rigueur in this field – confocal microscopy of the condensates, partitioning of components between condensate and bulk, FRAP recovery times, apparent viscosity, and client partitioning.
- (ii) Since NPM1-RNA is a complex condensate system the authors break down this phenomenon by looking at glycine effects on ‘uni-dimensional’ model condensate systems. The distinction between the electrostatic- and cation/π-π- driven condensates is established.
- (iii) These effects are extended to other amino acids, which offer the additional contribution of aa-sidechain interactions.
- (iv) General conclusions from the above highlight the molecular basis for amino-acid interactions with biomolecular condensates. These carry important implications for understanding biological condensates/cellular compartmentalization as well as designing tailor-made systems for other functions.

The effects of small (often bio-related) molecules upon the behavior of polypeptides in aqueous environment is well-known and extensively studied. The effects of urea, glycerol, arginine, glutamate and PEG to name a few on protein folding, assembly and aggregation has been thoroughly investigated. Such effects are a combination of two factors, (i) competition of the introduced solute with polypeptide groups on their interactions within the polypeptides and/or with the solvent, (ii) interactions of the solute with the solvent itself, thus modulating its availability for polypeptide-solvent interactions. In this sense it is not surprising that the zwitterionic and highly-soluble glycine, particularly at ~1 M concentrations, should exert such a modulating effect on the condensate systems studied here. Also, the difference between electrostatically-guided and hydrophobically-guided condensation systems is consistent with the nature of glycine and other amino acids. Nevertheless, it is of utmost importance to methodically uncover the molecular origins of these effects. From this standpoint the work described here is an important contribution, first establishing an effect on the original biological condensate and then simplifying the study by looking at artificial systems in order to deconvolute the various interactions and their contributions to the observed effect.

In light of the methodical approach to these basic science questions, and the current interest in biocondensates from both a biological and a biomaterials vantage point, I feel this MS can be improved to be worthy of publication in NatComm. I do, however, identify several points which need to be addressed in a serious revision in order to make this study more comprehensive and complete:

- (i) The effects of glycine (p.7) on NPM1 concentration in the dilute phase and NMR spectral changes are fitted to a Langmuir isotherm. I feel this is somewhat simplistic, inherently assuming glycine interferes with condensation only by protein/polypeptide binding. As mentioned above, it may also compete for solvent, rendering the solvent less available for interactions with components of the condensate system. At 0.5-2.0 M concentrations the amino acid comprises close to 2% of the bulk solvent and is not just a ‘solute’ in the regular sense. Have other models been tried, compared to the Langmuir model, and rejected? One suggestion would be defining an effective concentration of polypeptides which increases with increasing Gly concentration and decreasing solvent availability. I would expect this point to be addressed in more detail.
- (ii) On the same subject as (i), Figure 3a shows a Langmuir fitting for data in the 0-0.9 M range resulting in a K_d of 0.9 M, meaning that concentrations above the K_d were not sampled. This seems unreasonable – for example, a biphasic behavior (emanating from a solvent effect) would not have been picked up. Concentrations of 1.5-2 times K_d are required to get a confident affinity estimate.
- (iii) The relation between the various K_d values (lines 225-230) is unclear, the difference between K_d and “overall K_d” needs to be better explained.
- (iv) The effects of glycine on pH of condensation in the FFsFF system (Figure 2C) are not very high, 0.2-0.4 pH units between 0 and 1 M glycine. The authors should consider how the pH affects the pK_a's of the ionizable groups of the polypeptides (since this may be an alternative explanation for the results). Also, high glycine concentrations may directly affect the pH of the solution; there is no mention of how the pH was maintained, or whether it was remeasured after each incremental addition of glycine.
- (v) The amino acid glutamate is found to be unique in its effects, but this is not justified mechanistically, nor is any experiment suggested to clarify this point. Can the authors at least speculate or draw support from the literature as to the significance of this finding?

I also posit a comment on the format of the MS: The header of the main section reads “Results and Discussion”, and later there is a “Discussion” section, which would appear to be an error. However, the “Discussion” actually describes additional experiments employing two glycine similes, betaine and taurine, as well as additional condensation systems. This seems like an additional sub-section of the Results, not part of a Discussion. The authors should consider revising the MS structure

here.

In summary, this is a promising study which addresses a true need and will be of great interest to the readership. Once these issues are handled the manuscript will be a candidate for publication.

Reviewer #3

(Remarks to the Author)

MAJOR COMMENTS

Line 53. The authors should further emphasize the biological relevance of their work, particularly in relation to the selected model condensates. While protein synthesis and amino acid availability are classically associated with the cytoplasm, evidence suggests that protein synthesis may also occur in the nucleus (Mazia D., Nature 175, 1955). This could potentially support the rationale for studying AAs interactions in nucleolar-like condensates (NPM1-RNA). However, this context is currently missing from both the Introduction and Discussion. Moreover, the Introduction appears to set the focus on nucleoli (NPM1-RNA model), yet the study includes a variety of model condensates that are structurally and functionally distant from nucleolar systems. Without a clear rationale, the broader aim of the study becomes unclear. If the goal is to investigate the role of AAs in a range of biomolecular condensates, regardless of type, I encourage the authors to restructure the Introduction accordingly and better justify their choices of models.

Although every model helps to investigate a specific interaction type, it is not consistent with the nucleolus model.

For instance, the rationale behind the inclusion of ATP-containing condensates is not clearly presented. The authors should explain why free AAs would be expected to partition into these specific condensates.

Figure 1. The authors claim a difference in the fluorescence intensity of condensate with increasing glycine concentrations. For quantitative comparisons, it is essential that laser power and color bar scaling are consistent across all samples. Currently, information like laser power and color bars (for each image) is not provided.

Line 130. If free fluorescein is added to the solution as a viscosity tracer, clarification is needed on how the authors discriminate its fluorescence from that of NPM1-Alexa488. How can free Alexa488 partition within condensates? If this happens, the NPM1-Alexa488 is not appropriate to perform the quantification analysis on fluorescence images in Figure 1. When the authors comment on the diminution of the fluorescence intensity within condensates, the fluorescence should only derive from NPM1-Alexa488. This is critical, as any significant contribution from free dye would compromise the accuracy of the analysis.

Importantly, no RICS data (e.g., autocorrelation curves, retrieved diffusion coefficient) are included in the Supplementary. An example of the fluorescence images used for RICS analysis should also be shown.

Line 143. Could the authors please comment on the GFP size relative to the K72 peptide? While GFP is classified as a non-phase-separating protein, and it should not affect the LLPS of the K72 peptide, its size relative to K72 could affect the dynamics or partitioning of the peptide.

Line 147. In Figure 1, the authors measure the concentration of NPM1-Alexa488 from fluorescence images, starting from a calibration curve shown in S18. It seems they apply the same strategy for K72-GFP, but no calibration curve is shown. If fluorescence quantification is based on such calibration, a corresponding curve for K72-GFP should be included.

Additionally, please clarify whether the NPM1 calibration (Tecan) was performed in the same buffer and multi-well format as the experimental system. For consistency, fluorescence calibration derived directly from intensity images (as in Dada S.T., PNAS, 2023) might be more appropriate. The authors should briefly justify their methodological choices or rely on the same technique.

Figure 3, S1, S2, 5a, 5b. I strongly encourage the authors to include raw data (e.g., individual data points) in all bar plots.

Line 177. The experimental pH of 6.5 is below physiological levels. While some cellular compartments (e.g., lysosomes) or biomolecular condensates (Ausserwoger, biorXiv, 2024) do exhibit acidic pH, the authors should discuss how their chosen conditions relate to biological systems, particularly in the context of the FFsFF model. Similarly, Figure 2c (inlet) highlights results at pH 6.8. Why not emphasize data closer to physiological pH?

Figure S7 and 2c. The authors claim that turbidity increases (or FFsFF concentration in the dilute phase decreases) with increasing concentration of glycine (from 0 to 1 M) and pH (from 6 to 8). If the authors want to draw conclusions regarding pH-dependent effects of glycine concentration, the inclusion of error bars in the 3 curves is essential. Without them, it is difficult to assess whether observed differences reflect true trends or experimental variability.

Line 266. The rationale behind selecting glycine and proline (for the trimer) and only proline (for the octamer) for peptide experiments is not provided. The authors should briefly explain these choices and whether other amino acids were not considered.

Line 262. Can the authors further comment on the possible explanations for why E is the only AA promoting condensate formation? In the Discussion, the authors state that no specific AAs side chain is needed for interaction with condensate components. In the Results, E is highlighted as an outlier (since it promotes condensate formation). These two points appear to be at odds and should be reconciled.

Line 262. Glutamate (E) reportedly constitutes a significant portion (~40%) of the nuclear free amino acid pool in rats (Wang K.M., Nature 204, 1964). Given the fact that E is the only AA promoting condensate formation, the authors may reflect on its biological significance in relation to condensate formation.

Line 350 and 379. The use of 100 ng/μL RNA in NPM1-RNA condensates should be justified, particularly if the model aims to mimic nucleolar conditions. Is this concentration representative of in vivo RNA levels? If available, references to previously measured RNA concentrations in the nucleolus would be useful.

MINOR COMMENTS

Line 72. Ribosomal RNA is abbreviated as “RNA” throughout the manuscript, while here it is referred to as “rRNA”. For consistency, I recommend using a single abbreviation.

Figure 89. While the difference in circularity between 0 M and 0.9 M is visually evident, a more quantitative circularity analysis may help reveal potential differences also between intermediate conditions (e.g., 0.3 M and 0.9 M), which are difficult to discern from the images alone.

Figure 2. For the synthetic model condensates involving fluorescent molecules, the authors should consider including representative fluorescence images, similar to those shown in Figure 1. This would be particularly useful in Figure 2c, where visualizing changes in condensate formation across pH values could complement the quantitative data. Same for NPM1-RNA in the presence of RP3.

Figure S6. There is a typo in the x-axis label. Please correct it.

Figure S7. The pH values at which turbidity measurements were taken appear to differ between samples. Could the authors clarify the reasons behind this approach? For accurate comparison, it may be preferable to use the same number and the range of pH values across all samples. A similar issue is observed in Figure S16, where the Taurine sample has only one data point.

Line 60. Please verify reference 20. Lipinski W.P. et al. do not appear to mention FFsFF.

Line 178 and 179. Can the authors explain what they mean by “lower pH” and compared to what?

Figure 3b. K_d associated with S appears slightly lower than those for other amino acids. Could the authors comment on whether this difference is significant? Even in the absence of statistical tests, reporting means and associated errors would be helpful.

Line 205. LC-MS and NMR are introduced here without definition. While widely used, defining these acronyms upon first use would enhance accessibility for a broader scientific audience.

Line 284. The discussion of betaine and taurine results appears in the Discussion section. I recommend first presenting and commenting on these findings in the Results section before discussing them later. The same applies to PDDA and PAA condensates. Overall, a more detailed Discussion is needed.

Line 320. The authors refer to potential peptide-based strategies to counteract condensate aging and neurodegeneration. However, the link between neurodegenerative diseases and nucleolar function was not introduced earlier. Including some background in the Introduction would help contextualize this closing statement.

Line 349. The labeling procedures for NPM1, rRNA and K72 are not described.

Line 406. Can the authors comment on the equation or model used to calculate the apparent viscosity (e.g., Stokes-Einstein equation, ecc.)?

Figure S16, S18. Error bars are missing from individual data points in these figures.

Figure S18. The values on the y-axis are not fully visible. Please ensure all axis values are clearly visible.

Version 1:

Reviewer comments:

Reviewer #1

(Remarks to the Author)

In this revised manuscript titled "Amino acids bind to phase-separating proteins and modulate biomolecular condensate stability and dynamics", the authors have provided clarifications, conducted control experiments and have improved the overall findings by providing details. They have addressed previously raised issues regarding tie-line gradient analysis in Figure 1e and have updated the methods regarding fluorescent tags FITC/GFP. Additionally, the authors have performed control experiments to show poor preferential partitioning of amino acids into PDAA-PAA condensates using NMR spectroscopy as well as demonstrated using a sarcosine amino acid control, that the partitioning trends into WGR-4 are not correlated exclusively to pKa. In response to reviewer 2, the authors have also included in their response, MD simulations investigating the interactions between diproline/diglycine and proteins like lysozyme (from a different manuscript), indicating that indeed, local surface concentrations of dipeptides are higher than expected by a simple concentration effect, indicating sites of interaction. Although this is not direct evidence for amino acid interactions with biomolecular condensates, it suggests an interesting generality in the ideas communicated herein.

In light of these changes, I recommend the manuscript for publication in Nature Communications.

Reviewer #2

(Remarks to the Author)

I had several questions about the first submitted version of the manuscript, formulated in 4-5 main points. The main point was understanding the role of glycine as only binding to the condensate and overlooking potential effects on the solvent (environment).

In their reply the authors have presented convincing arguments why their interpretation is likely correct, and also provided proof of this in the form of results from a followup study (simulations).

Thus I consider this concern resolved.

Good answers including conducting complementary experiments as required have also addressed the other comments adequately.

In view of this I find the MS now worthy of publication in Nature Communications.

Reviewer #3

(Remarks to the Author)

The authors have made substantial revisions compared to the first version, particularly in explaining their rationale for using different condensate models and broadening the audience of their work. Most of my previous concerns have been adequately addressed. However, I still have a few remaining comments. To enhance reproducibility, I specifically recommend that the authors revisit the Methods section to ensure that all the experimental procedures are described in detail. Additionally, I believe that further improvements in clarity and consistency are needed to meet the standards for publication on Nature Communications. I leave the final decision on whether to accept the manuscript in its current form to the editor.

Line 44. In order to make the Introduction suitable for a more general public, it should be mentioned that stress granules are an example of BCs.

Line 59. At line 53 the authors state that they tested the effect of G on NPM1/RNA condensates, then at line 59 they mentioned other AAs ("all tested amino acids"). Although the message is clear after reading the manuscript, the reader might get lost here. I suggest rephrasing the Introduction.

Line 69. I would keep the results summarized here as general as possible. Thereby, I would avoid mentioning any Kd value, especially because the apparent dissociation constant is only introduced at line 199.

Figure 1. If the authors want to use "dilu" and "cond" to abbreviate "dilute" and "condensed", they should define these abbreviations first.

Figure S5, S9. I encourage the authors to add error bars or any confidence level to all curves. Same for data points in figures S8 and S14. When possible, please also indicate the number of measurements (N) or the number of independent experiments performed.

Figure S6. It seems that the internal organization of K10-D10 condensates is heterogeneous, in the sense that some brighter sub-compartments are visible. Is this expected?

Figure S7. Can the authors also include a representative image of K72-ATP condensates upon addition of 4 different AAs (figure S9) and the glycine derivatives (figure S16).

Figure 3b,c,d. Can the authors add the data points on top of the bar plots? Are the Kd values really oscillating between 0.1 and 1, or is it just a matter of data representation (e.g., bar plots instead of box plots)?

Line 183. The authors clarified that FFsFF concentration was determined by turbidity measurements. However, the procedure is not mentioned in the Methods section.

Line 196. Please add a reference to justify the binding isotherm.

Figure S17. I guess the authors measured the concentration of PDAA-PAA in the dilute phase in the same way as for NPM1 condensates and similar. Is PDAA or PAA fluorescently labeled? No details about the labeling or the sample preparation are mentioned in "condensate formation" (Methods).

Figure S20. Figure 1 shows that NPM1-RNA condensates have a diameter ranging approximately from 1 to 5 μm (according to the reported scale bar). Although the authors claim that the RICS frames are smaller than the condensates, Figure S20

shows a RICS frame that can be bigger than an NPM1 condensate (according to the scale bar). Additionally, are guest Alexa488 molecules preferentially partitioning into the condensates? If they are homogeneously distributed in the sample, then a brightfield image is crucial to understand if the measurement has been performed in the condensate. If not, the condensates should appear brighter than the dilute phase, and, if the nanomolar concentration of Alexa488 does not allow for a clear imaging, a brightfield is again required. To clarify this, I suggest showing either the bigger frame with one or more condensates and the “zoomed” field-of-view used for RICS measurements.

Line 221. The partition coefficient has been introduced at line 140. The abbreviation Kd should be reported the first time the partition coefficient is mentioned. This is also valid for other abbreviations (i.e., the one for AAs).

Line 372. What are A647 and A488 referring to? If A488 is referring to Alexa Fluor 488 (sometimes also mentioned as Alexa488 or Alexa 488), I encourage the authors to refer to it in a consistent way throughout the manuscript.

Line 377. Check the work “labelled”. K10 is labelled, not FITC.

Line 403. The authors are here describing the procedure to measure NPM1-A488 in the dilute phase, stating that the RNA is unlabelled. However, they also estimate the concentration of RNA with the same fluorescence-based analysis (line 408). They should adapt the paragraph by including both quantifications.

Line 427. I suggest moving the description of RICS measurements to the subsequent paragraph (“Diffusion coefficient measured by RICS”). Additionally, can the authors indicate the type of fitting model they used for RICS analysis and the autocorrelation curves?

Version 2:

Reviewer comments:

Reviewer #3

(Remarks to the Author)

In the revised manuscript, the authors have addressed my comments and suggestions, thus improving the overall consistency and reproducibility of the work. They have also properly answered the concerns raised by reviewers 1 and 2. I consider the manuscript now suitable for publication in Nature Communications.

► *Reviewer's comments are greyed out and italicized; authors' responses are in black font right below each group of comments, as well as highlighted in the manuscript in yellow.*

Reviewer #1 (Remarks to the Author):

In the article "Amino acids bind to phase-separating proteins and modulate biomolecular condensate stability and dynamics", Xu et al. investigate the modulation of coacervate properties in the presence of soluble proteogenic amino acids. Using a set of known biomolecular liquid-liquid phase separating systems that rely on varying non-covalent interactions such as electrostatics, cation/ π - π , they explore the differential role of amino acids such as glycine in controlling partitioning, network properties and material characteristics such as viscosity. These analyses are reliant on fluorescence from confocal microscopy, LC-MS data, turbidimetry as well as NMR spectroscopy on the studied in vitro systems. The role of backbone interactions with amino acids is highlighted, with a general trend observed for soluble amino acids and short repeat peptides that were investigated. The key findings indicate that high concentrations of soluble amino acids like glycine disfavor LLPS in case of heterotypic condensates (NPM1-RNA/K72-ATP/K10-D10) whereas they promote condensate formation for simple coacervates of peptides like FFsFF/WGR-4. Since amino acids and biomolecular condensates co-exist in vivo and it is important to understand and manipulate their materials properties, this work is of interest to the community as it augments existing understanding of the regulation of protein-protein interactions by amino acids. However, the authors should address the following concerns prior to this work being considered for publication.

► We appreciate the reviewer's positive assessment for our work. We have carefully considered all the comments and made substantial revisions to improve the clarity and overall quality of the manuscript. Below, we address each point in detail.

1) Results discussed in Figure 1 a-e, indicate that with increasing glycine concentration in case of the NPM1-RNA condensation, NPM1 solubilization in the dilute phase is favored. The authors add that RNA concentrations are not affected significantly (Figure S1). This would imply that in order to achieve effective charge complexation, some NPM1-RNA interactions would be substituted by zwitterionic glycine. This should change the coordination number z discussed in the tie-line gradient analysis on Page 3 and Figure 1 e. Additionally, why is the glycine not factored into this analysis either as a solvent modulator (0), or an additional solute (3) in spite of high concentration? Can the authors explain why they believe that all other variables in this analysis are constant so as to allow them to conclude that the increase in k is exclusively due to weaker associative interactions?

► That is a good point. As shown in **Figure 1e**, tie-line gradient k increases with increasing glycine concentration. This trend indicates that the effective interaction difference driving condensate formation (χ^A) decreases.

We agree that introducing glycine (solute 3) alters the coordination number z and the contact energies between solvent (water, component 0) and solutes (NPM1 as component 1, RNA as component 2). This makes it difficult to isolate glycine's effect on any specific interaction pair beyond its overall effect in weakening net interaction driving the condensate formation (χ^A).

To clarify it in the manuscript, we have revised the main text as follows:

“...by effective interaction difference for the condensate formation (χ^Δ): $k \approx -\frac{1}{(1+2\chi^\Delta)N_1\phi_1}$ where ϕ_1 and N_1 denote the volume fraction and length of component 1 (NPM1), respectively. $\chi^\Delta = \frac{z\Delta\mu}{k_B T}$, where $\Delta\mu$ denotes the contact energy difference between solvent-solute (water-NPM1, water-RNA, water-AA) pairs and the average of solvent-solvent and solute-solute interactions...As shown in **Figure 1e**, the tie-line gradient k increases with increasing glycine concentration, indicating a weaker net interaction driving the condensate formation.”

2) Alternatively, if the condensates are enriched in RNA when the glycine concentration is increased from 0 to 0.9 M, why is the partitioning of a basic peptide RP3 (Figure 1 h) not enhanced as the negatively charged RNA should favor partitioning of this cationic peptide?

► RP3 are arginine-rich peptides which are known to partition in NPM1-rRNA condensates due to electrostatic interaction with both NPM1 and RNA (*Musinova et al, Biochimica et Biophysica Acta (BBA)-Molecular Cell Research 2015, Yewdall et al, Biophysical Journal 2022*). In our system, the addition of glycine led to a clear decrease in NPM1 concentration within the condensates (**Figure 1**), while the RNA concentration remained relatively unchanged (**Figure S1**).

Given that RP3 interacts with NPM1, the reduced presence of NPM1 in condensates contributes to the observed decrease in RP3 partitioning. Another plausible contributing factor is that glycine may also weaken the RP3-NPM1 interaction. This could occur through glycine's ability to bind to amide groups present in both NPM1 and RP3, thereby interfering with their attractive interaction.

3) The electrostatic complexation driven condensates investigated in Figure 2 (K72-ATP and K10-D10) have fluorophores of vastly different molecular weights with the former tagged with GFP which is itself charged. In Figure 1, a 1:19 molar ratio for fluorescently tagged molecules is reported however it is unclear from the methods section on Condensate Formation, what the molar ratio of the labelled:unlabelled molecules used is for Figure 2 systems?

► Indeed, fluorescein isothiocyanate (FITC) has a molecular weight of 0.5 kDa while GFP has a molecular weight of 27 kDa. In addition, GFP has a negative net charge (-7).

In the systems shown in Figure 2, GFP-tagged K72, which is an elastin-like disordered polypeptide containing 72 lysine residues, was purified from *E. coli* and used directly to form condensates with ATP. Therefore, the labelled: unlabelled ratio for this system is 1:0. We note that the phase behaviour without amino acids of this system is well established and shown to be electrostatic interaction driven (Nakashima et al, *Nat Commun* 2021).

For K10–D10 condensates, FITC-labelled K10 (FITC-K10) was premixed with unlabelled K10 at a 1:49 molar ratio before mixing with unlabelled D10. The phase behavior of this system is also well established and shown to be charge-driven (Cakmak et al, *Nature Commun* 2020).

To improve clarity, we have revised the methods section on condensate formation as follows:

“K10-D10 condensates were prepared in HEPES buffer (final concentration 50 mM, pH 7.4), by adding D10 (final concentration 5 mM) to HEPES buffer followed by K10 (final concentration 5 mM, 1:49 molar ratio of fluorescein isothiocyanate (FITC) labelled).”

Although the fluorophores are of different sizes, both condensate model systems are shown to be driven primarily by electrostatic interactions, and both have a well-defined critical salt concentration. Our findings that both types of condensates show an increase in the saturation concentration of the condensate components in the dilute phase upon addition of amino acids, therefore support the conclusion that amino acids can weaken charge-driven condensation. The fact that we see a consistent weakening of the charge interactions in these condensates despite the vast size difference of the fluorophores in these systems makes the mechanism even more robust.

4) In Figure 3, using NMR and LC-MS data, the authors show preferential partitioning of amino acids into the complex coacervates of proteins/peptides. Also, in Figure S17 and Discussion (page 10), the authors show that amide groups in backbones are important for interactions which are absent in the PDDA-PAA system. Do amino acids partition into the PDDA-PAA system in the absence of such backbone amide interactions?

► It is another valuable point raised by the reviewer. To address it, we have done additional experiments to determine if amino acids partition into PDDA-PAA condensates. By using NMR spectroscopy, we quantified the concentrations of three amino acids (Pro, Ser, and Ala) in both the dilute and condensate phases.

As shown in **Figure S17b**, the partition coefficients for these amino acids were all approximately 1 (Pro \sim 1.0, Ser \sim 1.5, Ala \sim 1.4), indicating minimal or no preferential partitioning into the condensate phase. This observation also aligns with the lack of a modulation effect of amino acids on this system, as shown in **Figure S17a**.

Accordingly, we have added these results to the manuscript:

“we employed synthetic condensates formed by the electrostatic interaction between poly(diallyldimethyl-ammonium chloride) (PDDA) and poly(acrylic acid) (PAA)...We also found that there was no preferential partitioning of AAs into the condensate phase (**Figure S17b**).

Figure S17b: The measured partition coefficients (K_p) by NMR for three representative AAs (proline, serine, and alanine) in the condensate phase of PDDA-PAA systems.”

5) *Chemical shift perturbations in Figure 4 are interesting, with strong evidence for glycine-protein/peptide interactions. Could the authors comment on the changes in line shape including broadening upon addition of 0.6 M G-d5?*

► Indeed, chemical shift perturbations (CSPs) are strong evidence for interactions. Peak broadening is often observed when molecules with fast exchange dynamics slow down, or non-exchanging molecules speed up. In the cases of FFsFF and WGR-4, the broadening is indicative of an extra exchange process other than H₂O-NH exchange upon the addition of glycine. The NHs may be engaged in the weak binding event with glycine, and due to the intermediate exchange rate between the two populations (bound and free), line shapes broaden (T₂ broadening).

It is also interesting for K72 that it goes from broad (i.e., very intermediate exchange) to a sharper peak upon the addition of glycine. This could be due to the rate of on/off binding of glycine to the protein being faster than H₂O-NH or internal motions, thus outcompeting the intermediate exchange rate process and moving it further into faster exchange.

Either way, both CSPs and line shape changes are indicative of a change to the proton exchange dynamics occurring at the protein backbone, which coincides with the binding of glycine to the proteins/peptides.

6) *The comparison of Glycine with Betaine and Taurine is unclear in terms of the providing evidence for concluding “that the primary amine group of AAs most likely binds to the proteins in BC components through hydrogen bonding” (Page 9). The sulfonic acid group of taurine is a stronger acid, and capable of participating in H-bonding more readily than the carboxylate group of glycine, likely explaining the strong increase in dilute phase concentration of NPM1. It would likely be insightful to test N-acetylated glycine and C-amidated glycine to decouple the role of charge and hydrogen bonding from each termini.*

► We thank the reviewer for bringing up this point. When comparing betaine and glycine, the key structural difference is their amine groups: glycine has a primary amine, while betaine contains a quaternary amine. Primary amines can participate in hydrogen bonding, while quaternary amines cannot. This difference likely explains why betaine does not exhibit a modulation effect on NPM1-RNA condensates, whereas glycine does. This comparison supports our conclusion that “...primary amine group of AAs most likely binds to the proteins in BC components through hydrogen bonding”.

In terms of the C-terminus in AAs, we chose taurine, which bears sulfonic acid groups. We agree with the reviewer that the sulfonic acid group is a stronger acid than the carboxylic acid group in glycine, which have an enhanced ability to engage in hydrogen bonding. This may explain its more significant modulation effect. However, as far as we know, there is no other molecule that is structurally similar to glycine, and is still zwitterionic like glycine, but has a weaker acid group than a carboxylic acid group.

Indeed, we also considered both N-acetylated glycine and C-amidated glycine, but the reason that we did not proceed with these two molecules is that they act like ions rather than zwitterions. Glycine has a pI of 6 (pK_a = 2.3 and 9.6) while N-acetylated glycine has a pK_a/pI of 3.7 and C-amidated glycine has a pK_a/pI > 10. As a result, these compounds have a strong charge-screening effect on the GFP-K72/ATP and K10/D10 condensate systems, similar to NaCl. By contrast, zwitterionic species hardly increase the ionic strength of the medium, and we can study their influence on protein backbone interactions at concentrations up to 1 M without disrupting the ion pairs.

Altogether, we remain confident that our control experiments with betaine are sufficient to support the conclusion that the primary amine plays an important role in the condensate modulation we

observed. On the other hand, it is difficult to conclude that the carboxylic acid group plays no role, since the sulfonic acid in taurine is indeed a stronger acid, which likely explains why the dilute phase concentration increases more significantly upon addition of taurine. Nevertheless, if the carboxylic acid group could engage in backbone interactions, we would expect to see a change in dilute phase concentration upon addition of betaine.

7) Figure S14 shows the anomalous trend with proline for WGR-4 partitioning. Could this possibly be due to the higher pKa for the cyclic NH vs all the free NH₂ termini other in amino acids?

► That is indeed a very intriguing hypothesis. We have performed an experiment to test this hypothesis by examining the effect of sarcosine (Sac), which bears a methylated (secondary) amine group with a pKa of 11.64, higher than the pKa of primary amines. For comparison, proline's (Pro) cyclic secondary amine has a pKa of 10.64.

As shown in the results below, unlike Pro, which promotes condensate dissolution, Sac exhibited a clear condensate-promoting effect, consistent with the effect of other amino acids reported in **Figure S14**. This contrast highlights that the difference in modulation effects is not solely due to pKa.

Figure 1: The effects of proline (Pro) and sarcosine (Sac) on a) WGR-4 concentration in the dilute phase and b) the turbidity of the whole solution for the WGR-4 system.

Additionally, the manuscript can benefit from a few minor edits –
1) On Page 5, the line “In agreement with FRAP...” should end with a reference to Figure 1g.

► We have added the reference to **Figure 1g** to the sentence.

2) Figure S1 b is missing the concentration units on the Y-axis.

► We have added the concentration units on the Y-axis in Figure S1b.

3) Figure S6 X-axis label has a typo for concentration.

► We have corrected the typo.

4) Figure 6 and Figure S16: Glycine should be represented in zwitterionic form for consistency with betaine.

► We have changed all the AA structures into zwitterionic forms for the sake of consistency.

Reviewer #2 (Remarks to the Author):

The MS titled “Amino acids bind to phase-separating proteins and modulate biomolecular condensate stability and dynamics” focuses on the timely and highly interesting phenomenon of biomolecular assemblies and aims to dissect and analyze the ways in which this behavior is modulated by amino acids. Briefly, the study addresses the following:

(i) The ‘biological’ NPM1-RNA condensate system is shown to be modulated by high concentrations of glycine. This is established by an array of experiments that are de rigueur in this field – confocal microscopy of the condensates, partitioning of components between condensate and bulk, FRAP recovery times, apparent viscosity, and client partitioning.

(ii) Since NPM1-RNA is a complex condensate system the authors break down this phenomenon by looking at glycine effects on ‘uni-dimensional’ model condensate systems. The distinction between the electrostatic- and cation/pi-pi- driven condensates is established.

(iii) These effects are extended to other amino acids, which offer the additional contribution of aa-sidechain interactions.

(iv) General conclusions from the above highlight the molecular basis for amino-acid interactions with biomolecular condensates. These carry important implications for understanding biological condensates/cellular compartmentalization as well as designing tailor-made systems for other functions.

The effects of small (often bio-related) molecules upon the behavior of polypeptides in aqueous environment is well-known and extensively studied. The effects of urea, glycerol, arginine, glutamate and PEG to name a few on protein folding, assembly and aggregation has been thoroughly investigated. Such effects are a combination of two factors, (i) competition of the introduced solute with polypeptide groups on their interactions within the polypeptides and/or with the solvent, (ii) interactions of the solute with the solvent itself, thus modulating its availability for polypeptide-solvent interactions. In this sense it is not surprising that the zwitterionic and highly-soluble glycine, particularly at ~1 M concentrations, should exert such a modulating effect on the condensate systems studied here. Also, the difference between electrostatically-guided and hydrophobically-guided condensation systems is consistent with the nature of glycine and other amino acids. Nevertheless, it is of utmost importance to methodically uncover the molecular origins of these effects. From this standpoint the work described here is an important contribution, first establishing an effect on the original biological condensate and then simplifying the study by looking at artificial systems in order to deconvolute the various interactions and their contributions to the observed effect. In light of the methodical approach to these basic science questions, and the current interest in biocondensates from both a biological and a biomaterials vantage point, I feel this MS can be improved to be worthy of publication in NatComm. I do, however, identify several points which need to be addressed in a serious revision in order to make this study more comprehensive and complete:

► We thank the reviewer for the thoughtful and constructive comments. We have addressed each of them carefully and thoroughly below.

(i) The effects of glycine (p.7) on NPM1 concentration in the dilute phase and NMR spectral changes are fitted to a Langmuir isotherm. I feel this is somewhat simplistic, inherently assuming glycine interferes with condensation only by protein/polypeptide binding. As mentioned above, it may also compete for solvent, rendering the solvent less available for interactions with components of the condensate system. At 0.5-2.0 M concentrations the amino acid comprises close to 2% of the bulk solvent and is not just a 'solute' in the regular sense. Have other models been tried, compared to the Langmuir model, and rejected? One suggestion would be defining an effective concentration of polypeptides which increases with increasing Gly concentration and decreasing solvent availability. I would expect this point to be addressed in more detail.

► This is an excellent point raised by the reviewer. We note that our NMR experimental data, shown in **Figure 4**, indicate that glycine binds to the protein backbones, as evidenced by the observed chemical shift perturbations, rather than competing for solvent. Moreover, if glycine were to compete for solvent, the formation of condensates composed of polymers without amide backbones, such as PDDA/PAA, should also be enhanced. By contrast, we find that PDDA/PAA condensates are unaffected by the addition of similar concentrations of glycine.

To further support this finding, we have also conducted molecular dynamics (MD) simulations, which directly confirm the binding interactions between amino acids and proteins. These simulation results are included as a part of a separate manuscript currently under review, and the relevant parts are copied below for convenience.

Finally, we note that the fits to a Langmuir isotherm in our manuscript are not intended to obtain a precise value for the dissociation constant, but rather an estimate of the order of magnitude of the effective affinity between the amino acids and protein. Since our aim is not to find and quantitatively compare the precise dissociation constants for amino acids to different proteins, we did not test other binding isotherms. Nevertheless, the assumption in the Langmuir binding isotherm all seems reasonable for the case of glycine binding to polypeptide/protein backbones: 1) there is a finite number of binding sites, namely the amide groups in the protein backbone; 2) there is no interaction between binding sites, as the amide groups are separated by an amino acid and side chains; 3) the binding is reversible, as it is likely based on hydrogen bonding. We have also added a sentence to our manuscript to emphasize that the fits are used only to estimate the affinity between the amino acids and protein backbones.

In addition, the suggested alternative to define an effective polypeptide concentration, which increases with glycine concentration, would not be able to explain the difference between the NPM1/RNA, GFP-K72/ATP and K10/D10 condensate systems on the one hand, and the WGR-4 and FFsFF condensates on the other hand because an increased effective polypeptide concentration would always enhance condensate formation, regardless of the chemical nature of driving force.

[Redacted]

[Redacted]

[Redacted]

(ii) On the same subject as (i), Figure 3a shows a Langmuir fitting for data in the 0-0.9 M range resulting in a K_d of 0.9 M, meaning that concentrations above the K_d were not sampled. This seems unreasonable – for example, a biphasic behavior (emanating from a solvent effect) would not have been picked up. Concentrations of 1.5-2 times K_d are required to get a confident affinity estimate.

► Thank you for the helpful suggestion. To obtain a more reliable estimation of the dissociation constant (K_d), we repeated the experiment using a broader range of glycine concentrations, extending up to 1.98 M. This improved data range led to a fitted K_d value of 1.0 ± 0.2 M, as shown in the revised Figure 3a below. We have also replaced the original plot and updated the related text in the manuscript.

Figure 3a: The NPM1 protein concentration in the dilute phase after adding glycine at varying concentrations up to 1.98 M and the fitting curve (black) with 95% confidence band (red) using the Langmuir-type binding model.

(iii) The relation between the various K_d values (lines 225-230) is unclear, the difference between K_d and "overall K_d " needs to be better explained.

► Thanks for pointing it out. In this case, we can measure binding of the glycine to both the backbone next to Glycine residues and next to Valine residues. The individual K_d for G-d5 binding to amide groups in G and V residues was estimated to be 1.5 ± 0.2 M and 1.7 ± 0.3 M, respectively (Figure 4d). Since both binding sites are independent and their dissociation constants are similar within experimental error, we can estimate an overall effective dissociation constant K_d as: $K_{d(\text{overall})} = \frac{c(\text{G})c(\text{K72})}{c(\text{K72-bound to G})+c(\text{K72-bound to V})}$

This leads to the relation:

$$\frac{1}{K_{d(\text{overall})}} = \frac{1}{K_{d1}} + \frac{1}{K_{d2}}$$

We have accordingly added this explanation in the main text for improved clarity.

(iv) The effects of glycine on pH of condensation in the FFsFF system (Figure 2C) are not very high, 0.2-0.4 pH units between 0 and 1 M glycine. The authors should consider how the pH effects the pKa's of the ionizable groups of the polypeptides (since this may be an alternative explanation for the results). Also, high glycine concentrations may directly affect the pH of the solution; there is no mention of how the pH was maintained, or whether it was remeasured after each incremental addition of glycine.

► Thanks for the insightful question. Regarding the pH changes following glycine addition, indeed, the pH varied slightly with increasing glycine concentration. We measured the pH after each addition of glycine to ensure accuracy. The observed pH values are summarized in the table below, which is also included in the SI for clarity, as **Table S1**. The pH values in **Figure 2c** and **S7** were already corrected for these pH changes.

As can be seen, the addition of glycine typically decreased the pH, which leads to a destabilization of the FFsFF condensates, as has been reported in earlier work (Abbas et al, Nature Chem. 2021). By contrast, we find here that the addition of glycine leads to enhanced condensate formation, counteracting the effect of the slight pH decrease.

Moreover, the NMR results in **Figure 4b** directly show that glycine binds to the backbone amide groups of the peptide and the aromatic side groups.

Table S1: pH measurement of Britton–Robinson (BR) buffer for FFsFF after the addition of glycine.

pH of buffer	pH after addition of 0.3 M G	pH after addition of 1 M G
6.23	6.13	6.00
6.53	6.44	6.29
6.78	6.69	6.53
7.08	6.97	6.79
7.37	7.25	7.04
7.69	7.50	7.26
7.99	7.71	7.43
8.36	7.94	7.62

(v) The amino acid glutamate is found to be unique in its effects, but this is not justified mechanistically, nor is any experiment suggested to clarify this point. Can the authors at least speculate or draw support from the literature as to the significance of this finding?

► We thank the reviewer for raising this point. Indeed, glutamate (E) has been identified in several studies as a special amino acid that promotes the formation of biomolecular condensates, which aligns with our findings. The examples include:

- 1) LLPS of FET family proteins—FUS, EWSR1, and TAF15 (Kar et al., Nat. Commun., 15, 4408, 2024);
- 2) LLPS and DNA-binding cooperativity of the E. coli single-stranded DNA-binding protein (Kozlov et al., J. Mol. Biol., 434, 9, 2022);
- 3) LLPS of the cell division protein FtsZ from E. coli (Paccione et al., Biochemistry, 61, 22, 2022).

Additionally, both Kar et al. and Kozlov et al. showed that glutamate is preferentially excluded from peptide backbones and side chains, thus enhancing protein–protein association and promoting condensate formation.

To reflect this, we have added the following sentences to the main text:

“Similarly, except for glutamate (E), all the neutral AAs tested show a condensate dissolution effect on K72-ATP system. The enhanced condensate formation by the addition of E is likely due to its preferential exclusion from peptide backbones and side chains, thereby promoting protein–protein association and stabilizing the condensate phase (Kar et al., Nat. Commun., 15, 4408, 2024; Kozlov et al., J. Mol. Biol., 434, 9, 2022). E may be exploited by cells to promote phase separation of specific proteins, potentially as a mechanism to regulate condensate formation in vivo (Kar et al., Nat. Commun., 15, 4408, 2024; Kozlov et al., J. Mol. Biol., 434, 9, 2022, Paccione et al., Biochemistry, 61, 22, 2022)”

I also posit a comment on the format of the MS: The header of the main section reads “Results and Discussion”, and later there is a “Discussion” section, which would appear to be an error. However, the “Discussion” actually describes additional experiments employing two glycine similes, betaine and taurine, as well as additional condensation systems. This seems like an additional sub-section of the Results, not part of a Discussion. The authors should consider revising the MS structure here.

► We thank the reviewer for pointing this out. We have changed the second “Discussion” into “Towards a molecular understanding of amino acid-mediated condensate modulation” as a sub-section in “Results and Discussion” section.

In summary, this is a promising study which addresses a true need and will be of great interest to the readership. Once these issues are handled the manuscript will be a candidate for publication.

► We thank the reviewer for their encouraging remarks and share their enthusiasm for the study. We have made substantial revisions in response to each comment to enhance the quality of the manuscript.

Reviewer #3 (Remarks to the Author):

MAJOR COMMENTS:

Line 53. The authors should further emphasize the biological relevance of their work, particularly in relation to the selected model condensates. While protein synthesis and amino acid availability are classically associated with the cytoplasm, evidence suggests that protein synthesis may also occur in the nucleus (Mazia D., Nature 175, 1955). This could potentially support the rationale for studying AAs interactions in nucleolar-like condensates (NPM1-RNA). However, this context is currently missing from both the Introduction and Discussion. Moreover, the Introduction appears to set the focus on nucleoli (NPM1-RNA model), yet the study includes a variety of model condensates that are structurally and functionally distant from nucleolar systems. Without a clear rationale, the broader aim of the study becomes unclear. If the goal is to investigate the role of AAs in a range of biomolecular condensates, regardless of type, I encourage the authors to restructure the Introduction accordingly and better justify their choices of models. Although every model helps to investigate a specific interaction type, it is not consistent with the nucleolus model. For instance, the rationale behind the inclusion of ATP-containing condensates is not clearly presented. The authors should explain why free AAs would be expected to partition into these specific condensates.

► We thank the reviewer for the thoughtful suggestions aimed at broadening the context and improving the logical flow of the manuscript. In response, we have revised the Introduction to more clearly articulate the rationale behind studying amino acids in the context of nucleolar-like condensates and model systems with distinct interaction types. The following changes were made:

To support the rationale for studying amino acids in nucleolar-like condensates (NPM1–RNA), we have added the following sentences to Line 52:

“Indeed, AAs are abundant within the nucleus (Errera et al., Biochim. Biophys. Acta, 49(1), 1961), and have been implicated in the regulation of nucleolar function and nuclear DNA replication (Bailey et al., PNAS, 73(9), 1976).”

To justify our use of a range of model condensates, the following explanation has been added to Line 56:

“NPM1–RNA condensates are formed through a combination of complex interactions (Gallo et al., J. Biol. Chem., 287(32), 2012). To disentangle the role of specific interaction types in condensate behaviour, we examined four additional model systems driven by different singular dominant driving forces: K72/ATP29, polyLys/polyAsp (K10/D10)30 (both formed by electrostatic interaction), FFsFF31 (formed by π - π stacking) and WGR-4 peptide32 (formed mainly by cation- π interactions).”

To clarify the rationale for testing AA partitioning, we added the following to Line 49:

“Given that AAs have been reported to modulate various biomolecular condensate systems, we hypothesize that free AAs may interact with condensate components, partition into condensates, thereby influencing their stability and dynamic properties.”

We believe these additions strengthen the manuscript by providing a more comprehensive and logically structured framework for the study.

Figure 1. The authors claim a difference in the fluorescence intensity of condensate with increasing glycine concentrations. For quantitative comparisons, it is essential that laser power and color bar scaling are consistent across all samples. Currently, information like laser power and color bars (for each image) is not provided.

► Thank you for pointing that out. The laser power was set to 50%, and the excitation wavelength was 485 nm at 25% intensity. These settings were kept constant across all microscopy sessions to ensure the comparability of the data.

Accordingly, we have now added the color bar scale and laser power information to the caption of revised **Figure 1**, as shown below.

Line 130. If free fluorescein is added to the solution as a viscosity tracer, clarification is needed on how the authors discriminate its fluorescence from that of NMP1-Alexa488. How can free Alexa488 partition within condensates? If this happens, the NMP1-Alexa488 is not appropriate to perform the quantification analysis on fluorescence images in Figure 1. When the authors comment on the diminution of the fluorescence intensity within condensates, the fluorescence should only derive from NMP1-Alexa488. This is critical, as any significant contribution from free dye would compromise the accuracy of the analysis. Importantly, no RICS data (e.g., autocorrelation curves, retrieved diffusion coefficient) are included in the Supplementary. An example of the fluorescence images used for RICS analysis should also be shown.

► In the RICS experiment, unlabelled NPM1 proteins were used. Alexa488 was used as a freely diffusing guest molecule, which spontaneously partitions into the condensates. As a result, all observed fluorescence originates exclusively from the guest Alexa488 molecules. To ensure that the fluorescent probe did not alter the properties of the condensates, its concentration within the condensates was kept in the nanomolar range.

Below, we present an example of the RICS analysis, including a representative image frame used for analysis (**Figure S20**), the measured autocorrelation curve (**Figure S21a**), and the fitted autocorrelation curve (**Figure S21b**). This particular fit yielded a diffusion coefficient of $48 \mu\text{m}^2/\text{s}$. It is important to note that, in RICS, the analysis is performed by closely zooming in on a localized region within a single condensate, using a pixel size of 20 nm. As a result, the edges of the droplet are not visible in the example image (**Figure S20**) because the focus is on assessing local diffusion behaviour within the droplet.

Accordingly, we have added all these data in the SI (**Figure S20 and S21**).

Figure S20: An example frame used for RICS analysis.

Figure S21: (a) the measured autocorrelation curve and (b) the fitted autocorrelation curve for Figure 3. The fitting yielded a diffusion coefficient of $48 \mu\text{m}^2/\text{s}$.

In addition, the raw diffusion coefficient data, as shown below, have also been included in the SI (**Figure S3b**).

Figure S3b: Diffusion coefficient (D) of fluorescein (Alexa Fluor 488) in NPM1-RNA condensates after the addition of glycine (0, 0.3, and 0.9 M), calculated by fitting the autocorrelation curves of Alexa488.

Line 143. Could the authors please comment on the GFP size relative to the K72 peptide? While GFP is classified as a non-phase-separating protein, and it should not affect the LLPS of the K72 peptide, its size relative to K72 could affect the dynamics or partitioning of the peptide.

► We thank the reviewer for the comment regarding the potential influence of GFP tagging on K72 phase behaviour. GFP has a molecular weight of approximately 27 kDa, while the K72 peptide is approximately 38 kDa. We note that the phase behavior of GFP-K72/ATP condensates has been extensively studied and is described in detail in earlier work (Nakashima et al, Nat Commun 2021). Here, we specifically look at how this phase behavior changes as a result of the addition of amino acids, which is studied with the same GFP-K72 polypeptide with the same size ratio of GFP and K72.

The GFP tag was used for visualization and quantification purposes, enabling fluorescence microscopy imaging and accurate concentration measurements of K72. GFP-K72 fusion constructs are widely employed in LLPS research and have been validated in multiple studies (Ma et al., Nat. Commun., 12, 3613, 2021; Nakashima et al., Nat. Commun., 12, 3819, 2021; Te et al., Nat. Nanotechnol., 13(9), 849–855, 2018).

In our study, GFP-tagged K72 (GFP-K72) was directly expressed in cells and used exclusively to form condensates with ATP, consistent with prior work (e.g., Nakashima et al., Nat. Commun., 2021). Since no unlabelled K72 was used, there is no heterogeneity in labelling, and thus the dynamics and partitioning of GFP-K72 are consistent in our study.

Line 147. In Figure 1, the authors measure the concentration of NMP1-Alexa488 from fluorescence images, starting from a calibration curve shown in S18. It seems they apply the same strategy for K72-GFP, but no calibration curve is shown. If fluorescence quantification is based on such calibration, a corresponding curve for K72-GFP should be included. Additionally, please clarify whether the NPM1 calibration (Tecan) was performed in the same buffer and multi-well format as the experimental system. For consistency, fluorescence calibration derived directly from intensity images (as in Dada S.T., PNAS, 2023) might be more appropriate. The authors should briefly justify their methodological choices or rely on the same technique.

► The calibration curve for GFP-K72 has been included in **Figure S19**.

Figure S29: Calibration curves for calculating the GFP-K72 concentrations in the dilute phase by fluorescence intensity measurement in the plate reader (Tecan Spark M10, 485/535 nm for GFP-K72). The fitting equation is also displayed alongside the fitted calibration curve.

To ensure accurate quantification, the calibration was performed using a known concentration series of NPM1/NPM1-A488 (1:19 molar ratio labelled). These measurements were conducted in the same buffer conditions and multi-well plates as those used in the experimental system.

For clarity, the following statement has been added to the Methods section:

“Calibration curves were performed using a series of known concentrations of NPM1/NPM1-A488 (1:19 molar ratio labelled) prepared in the same buffer and multi-well plates as the experimental samples.”

Figure 3, S1, S2, 5a, 5b. I strongly encourage the authors to include raw data (e.g., individual data points) in all bar plots.

► We thank the reviewer for pointing this out. Accordingly, we have revised Figures 3, S1, S2, 5a, 5b, S10 and S11 to include the raw data points.

Line 177. The experimental pH of 6.5 is below physiological levels. While some cellular compartments (e.g., lysosomes) or biomolecular condensates (Ausserwoger, biorXiv, 2024) do exhibit acidic pH, the authors should discuss how their chosen conditions relate to biological systems, particularly in the context of the FFsFF model. Similarly, Figure 2c (inlet) highlights results at pH 6.8. Why not emphasize data closer to physiological pH?

► FFsFF is fully soluble in water at pH values below 6, primarily due to electrostatic repulsion between charged FFsFF at acidic pH. However, as the pH increases above 6, the solubility decreases and condensate formation begins, driven by π - π stacking interactions as the charges are neutralized (Abbas et al., Nature Chemistry, 13(11), 2021: 1046–1054). Thus, the phase behavior of FFsFF is highly pH-dependent. We specifically chose to examine the pH range from 6.0 to 8.0 to capture this transition.

The reason for emphasizing pH 6.8 is that it lies near the midpoint of the sharp solubility-to-condensation transition, making it the most suitable pH value for studying the modulation of condensate formation with maximal sensitivity.

Figure S7 and 2c. The authors claim that turbidity increases (or FFsFF concentration in the dilute phase decreases) with increasing concentration of glycine (from 0 to 1 M) and pH (from 6 to 8). If the authors want to draw conclusions regarding pH-dependent effects of glycine concentration, the inclusion of error bars in the 3 curves is essential. Without them, it is difficult to assess whether observed differences reflect true trends or experimental variability.

► As suggested by the reviewer, we repeated the experiments two more times to verify the observed trend. Accordingly, **Figure 2c** has been updated with the new data, and the replicate results are now included in **Figure S7**:

Figure S7: Three replicates of the FFsFF concentration in the dilute phase and the turbidity of FFsFF at pH from 6 to 8 at different glycine concentrations (0, 0.3, and 1 M).

Figure 3c: FFsFF system at pH 6.8 (the FFsFF concentration in the dilute phase and the turbidity of the whole solution at pH from 6 to 8 shown in **Figure S7**).

Created in BioRender. Stellacci, F. (2025) <https://BioRender.com/p727lpy>

Line 266. The rationale behind selecting glycine and proline (for the trimer) and only proline (for the octamer) for peptide experiments is not provided. The authors should briefly explain these choices and whether other amino acids were not considered.

► Glycine and proline were selected for oligomer (trimer and octamer) experiments due to their high solubility, which was expected to extend to their respective oligomers. However, while the glycine trimer (G_3) and proline trimer (P_3) are very soluble, the glycine octamer (G_8) exhibited very poor solubility in water. As a result, data for (G_8) could not be obtained.

To clarify this, we have added the following explanations to the main text:

“Furthermore, two short peptides, a glycine trimer (G_3) and proline trimer (P_3), were employed to investigate if they could also modulate condensate formation as both glycine and proline are among the most water-soluble AAs.”

“Moreover, the linear transferability in the modulation effect is still true for proline octamer (P_8), whereas data for the glycine octamer (G_8) could not be obtained due to its very low solubility.”

Line 262. Can the authors further comment on the possible explanations for why E is the only AA promoting condensate formation? In the Discussion, the authors state that no specific AAs side chain is needed for interaction with condensate components. In the Results, E is highlighted as an outlier (since it promotes condensate formation). These two points appear to be at odds and should be reconciled.

Line 262. Glutamate (E) reportedly constitutes a significant portion (~40%) of the nuclear free amino acid pool in rats (Wang K.M., *Nature* 204, 1964). Given the fact that E is the only AA promoting condensate formation, the authors may reflect on its biological significance in relation to condensate formation.

► We thank the reviewer for raising this point. Indeed, glutamate (E) has been identified in several studies as a special amino acid that promotes the formation of biomolecular condensates, which aligns with our findings. The examples include:

- 1) LLPS of FET family proteins—FUS, EWSR1, and TAF15 (Kar et al., *Nat. Commun.*, 15, 4408, 2024);
- 2) LLPS and DNA-binding cooperativity of the E. coli single-stranded DNA-binding protein (Kozlov et al., *J. Mol. Biol.*, 434, 9, 2022);
- 3) LLPS of the cell division protein FtsZ from E. coli (Paccione et al., *Biochemistry*, 61, 22, 2022).

These findings collectively suggest that cells may exploit glutamate to promote phase separation of specific proteins, potentially as a mechanism to regulate condensate formation *in vivo*.

Additionally, both Kar et al. and Kozlov et al. showed that glutamate is preferentially excluded from peptide backbones and side chains, thus enhancing protein–protein association and promoting condensate formation.

To reflect this, we have added the following sentences to the main text:

“Similarly, except for glutamate (E), all the neutral AAs tested show a condensate dissolution effect on K72-ATP system. The enhanced condensate formation by the addition of E is likely due to its preferential exclusion from peptide backbones and side chains, thereby promoting protein–protein association and stabilizing the condensate phase (Kar et al., *Nat. Commun.*, 15, 4408, 2024; Kozlov et al., *J. Mol. Biol.*, 434, 9, 2022). E may be exploited by cells to promote phase separation of specific proteins, potentially as a mechanism to regulate condensate formation *in vivo* (Kar et al., *Nat. Commun.*, 15, 4408, 2024; Kozlov et al., *J. Mol. Biol.*, 434, 9, 2022, Paccione et al., *Biochemistry*, 61, 22, 2022).”

Line 350 and 379. The use of 100 ng/μL RNA in NPM1-RNA condensates should be justified, particularly if the model aims to mimic nucleolar conditions. Is this concentration representative of *in vivo* RNA levels? If available, references to previously measured RNA concentrations in the nucleolus would be useful.

► The RNA concentration used in this study is not intended to represent *in vivo* RNA levels. Instead, it was chosen based on an established preparation protocol for NPM1–RNA condensates, as reported in the literature (Yewdall et al., *Biophysical Journal*, 121(20), 2022: 3962–3974).

MINOR COMMENTS:

Line 72. Ribosomal RNA is abbreviated as “RNA” throughout the manuscript, while here it is referred to as “rRNA”. For consistency, I recommend using a single abbreviation.

► We have changed it to “RNA” in the revised main text.

Figure 89. While the difference in circularity between 0 M and 0.9 M is visually evident, a more quantitative circularity analysis may help reveal potential differences also between intermediate conditions (e.g., 0.3 M and 0.9 M), which are difficult to discern from the images alone.

► We have calculated the circularity of the condensates in **Figure 1b** using the particle analysis function in FIJI, and the results have now been included in **Figure S1a**:

Figure S1a: Condensate circularity in Figure 1b, using the particle analysis function in FIJI.

Figure 2. For the synthetic model condensates involving fluorescent molecules, the authors should consider including representative fluorescence images, similar to those shown in Figure 1. This would be particularly useful in Figure 2c, where visualizing changes in condensate formation across pH values could complement the quantitative data. Same for NPM1-RNA in the presence of RP3.

► Thank the reviewer for the helpful suggestion. We have now added representative fluorescence images for the K72–ATP and K10–D10 systems to the SI (insets in **Figure S4** and **Figure S6** respectively).

Figure S4: Partition coefficient of K72 in the condensate phase for K72-ATP condensates as a function of glycine concentration and representative confocal fluorescence microscopy images of the condensates in K72 channel.

Figure S6: Partition coefficient of K10 in the condensate phase for K10-D10 condensates as a function of glycine concentration and representative confocal fluorescence microscopy images of the condensates in K10 channel.

For the FFsFF system shown in **Figure 2c**, no fluorescent probe was tagged with FFsFF. The peptide concentration in the dilute phase was quantified via absorbance at 256 nm, which is aromatic absorption peak of phenylalanyl residues. The condensate formation was estimated by measuring the turbidity of the solution (absorbance at 600 nm), following the method established by Abbas et al. (Nature Chemistry, 13(11), 2021: 1046–1054).

In addition, representative fluorescence images for RP3 in the NPM1–RNA condensates (**Figure 1h**) have also been included in the SI (**Figure S1d**).

Figure S1d: Confocal fluorescence microscopy images of NPM1-RNA condensates in RP3 channel after the addition of 0, 0.3, and 0.9 M glycine (laser power: 50%, the color bar on the left). Scale bar = 5 μm.

Figure S6. There is a typo in the x-axis label. Please correct it.

► We have corrected the typo.

Figure S7. The pH values at which turbidity measurements were taken appear to differ between samples. Could the authors clarify the reasons behind this approach? For accurate comparison, it may be preferable to use the same number and the range of pH values across all samples. A similar issue is observed in Figure S16, where the Taurine sample has only one data point.

► Regarding the pH value changes among the samples: The pH values varied slightly with the addition of glycine. To account for this, we measured the pH after each addition of glycine to ensure accuracy. The observed pH values are summarized in the table below, which is also included in the SI (**Table S1**) for clarity.

Table S2: pH measurement of Britton–Robinson (BR) buffer for FFsFF after the addition of glycine.

pH of buffer	pH after addition of 0.3 M G	pH after addition of 1 M G
6.23	6.13	6.00
6.53	6.44	6.29
6.78	6.69	6.53
7.08	6.97	6.79
7.37	7.25	7.04

7.69	7.50	7.26
7.99	7.71	7.43
8.36	7.94	7.62

Only a single concentration was tested for taurine (**Figure S16**) due to its limited solubility, which prevented testing at higher concentrations.

Line 60. Please verify reference 20. Lipinski W.P. et al. do not appear to mention FFsFF.

► Apologies for the confusion. We have now corrected the reference to the right one: Abbas et al., Nature Chemistry, 13(11), 2021: 1046–1054.

Line 178 and 179. Can the authors explain what they mean by “lower pH” and compared to what?

► Apologies for the confusion. We have now rewritten the sentence for clarity as follows:

“We found that with increasing glycine concentration, the concentration of FFsFF in the dilute phase decreased, and condensates formed at a lower pH, compared to the control without glycine addition (**Figure S7**).”

Figure 3b. K_d associated with S appears slightly lower than those for other amino acids. Could the authors comment on whether this difference is significant? Even in the absence of statistical tests, reporting means and associated errors would be helpful.

► To improve clarity, we have now included the fitted dissociation constants (K_d) and their standard deviations for the four amino acids in the caption of Figure 3, as follows:

“ $K_d = 0.6 \pm 0.2$ M for glycine, 1.1 ± 0.4 M for proline, 0.4 ± 0.1 M for serine, and 1.3 ± 0.6 M for alanine”.

While there are slight differences among these fitted values, we believe that these fitted K_d values are all of a similar order of magnitude (~ 1 M), and that our data do not permit to draw conclusion about significant differences between these amino acids.

Line 205. LC-MS and NMR are introduced here without definition. While widely used, defining these acronyms upon first use would enhance accessibility for a broader scientific audience.

► Thank the reviewer for the suggestion. We have now added the full names of the two techniques in the main text for clarity:

“liquid chromatography–mass spectrometry (LC–MS) and nuclear magnetic resonance spectroscopy (NMR).”

Line 284. The discussion of betaine and taurine results appears in the Discussion section. I recommend first presenting and commenting on these findings in the Results section before discussing them later. The same applies to PDDA and PAA condensates. Overall, a more detailed Discussion is needed.

► We have merged the results and discussion into a single unified section titled “Results and Discussion.” Therefore, the data from these experiments are described with the discussion presented immediately following the corresponding results for improved coherence.

Line 320. The authors refer to potential peptide-based strategies to counteract condensate aging and neurodegeneration. However, the link between neurodegenerative diseases and nucleolar function was not introduced earlier. Including some background in the Introduction would help contextualize this closing statement.

► We would like to clarify that the outlook of our study is not limited to nucleolar structures or functions. Rather, our broader aim is to use potential peptide-based strategies to target functional intracellular biomolecular condensates, which are implicated in neurodegenerative diseases.

To support this broader context, we have now cited the following four key review articles in the revised manuscript:

1. Deng, H., Gao, K., & Jankovic, J. *Nature Reviews Neurology*, 10, 337–348 (2014).
2. Han, T. W. et al. *Cell Chemical Biology*, 31(9), 1593–1609.
3. Alberti, S. & Hyman, A. A. *Nature Reviews Molecular Cell Biology*, 22, 196–213 (2021).
4. Shin, Y., & Brangwynne, C. P. *Science*, 357(6357), eaaf4382 (2017).

These references provide additional evidence for the relevance of ageing biomolecular condensates in the context of age-related neurological disorders.

Line 349. The labeling procedures for NPM1, rRNA and K72 are not described.

► Apologies for the oversight. The labeling procedures for NPM1, RNA, and K72 have now been included in the Methods section of the main text under the subsection:

“Labelling of NPM1, RNA and K72

NPM1-Alexa488 and RNA-Alexa647 labeling were performed as previously described by André, Yewdall, and Spruijt (*Biophysical Journal*, 122(2), 397–407, 2023).

GFP-labeled K72 was expressed and purified following the protocol reported by Nakashima et al. (*Nature Communications*, 12, 3819, 2021)”.

Line 406. Can the authors comment on the equation or model used to calculate the apparent viscosity (e.g., Stokes-Einstein equation, etc.)?

► The procedure for calculating the apparent viscosity of the condensates based on the diffusion coefficient of Alexa488 guest molecules in NPM1–RNA condensates has now been added to the Methods section:

“The apparent viscosity (η) of the condensates was calculated using the Stokes–Einstein relation, based on the measured diffusion coefficient (D) of Alexa488 and its hydrodynamic radius (R). The equation used is: $\eta = k_B T / 6\pi D R$, where k_B is the Boltzmann constant, T is the temperature, D is the diffusion coefficient, and R is the hydrodynamic radius of Alexa488, taken to be 1.4 nm.”

Figure S16, S18. Error bars are missing from individual data points in these figures.

► We have replotted **Figure S16** to include error bars based on replicate measurements.

For **Figure S18**, which shows the calibration curves for NPM1 and RNA, only a single set of concentration series for both NPM1 and RNA was measured for each curve. Therefore, no error bars are shown.

Figure S18. The values on the y-axis are not fully visible. Please ensure all axis values are clearly visible.

► We have replotted **Figure S18** to ensure the y-axis is visible.

► *Reviewer's comments are greyed out and italicized; authors' responses are in black font right below each group of comments, as well as highlighted in the manuscript in yellow.*

Reviewer #1 (Remarks to the Author):

In this revised manuscript titled "Amino acids bind to phase-separating proteins and modulate biomolecular condensate stability and dynamics", the authors have provided clarifications, conducted control experiments and have improved the overall findings by providing details. They have addressed previously raised issues regarding tie-line gradient analysis in Figure 1e and have updated the methods regarding fluorescent tags FITC/GFP. Additionally, the authors have performed control experiments to show poor preferential partitioning of amino acids into PDAA-PAA condensates using NMR spectroscopy as well as demonstrated using a sarcosine amino acid control, that the partitioning trends into WGR-4 are not correlated exclusively to pKa. In response to reviewer 2, the authors have also included in their response, MD simulations investigating the interactions between diproline/diglycine and proteins like lysozyme (from a different manuscript), indicating that indeed, local surface concentrations of dipeptides are higher than expected by a simple concentration effect, indicating sites of interaction. Although this is not direct evidence for amino acid interactions with biomolecular condensates, it suggests an interesting generality in the ideas communicated herein.

In light of these changes, I recommend the manuscript for publication in Nature Communications.

► We sincerely thank Reviewer 1 for acknowledging our revisions and recommending the manuscript for publication.

Reviewer #2 (Remarks to the Author):

I had several questions about the first submitted version of the manuscript, formulated in 4-5 main points. The main point was understanding the role of glycine as only binding to the condensate and overlooking potential effects on the solvent (environment).

In their reply the authors have presented convincing arguments why their interpretation is likely correct, and also provided proof of this in the form of results from a follow-up study (simulations).

Thus I consider this concern resolved.

Good answers including conducting complementary experiments as required have also addressed the other comments adequately.

In view of this I find the MS now worthy of publication in Nature Communications.

► We also sincerely appreciate Reviewer 2's recommendation for the publication of our work.

Reviewer #3 (Remarks to the Author):

The authors have made substantial revisions compared to the first version, particularly in explaining their rationale for using different condensate models and broadening the audience of their work. Most of my previous concerns have been adequately addressed. However, I still have a few remaining

comments. To enhance reproducibility, I specifically recommend that the authors revisit the Methods section to ensure that all the experimental procedures are described in detail. Additionally, I believe that further improvements in clarity and consistency are needed to meet the standards for publication on Nature Communications. I leave the final decision on whether to accept the manuscript in its current form to the editor.

► We thank Reviewer 3 for acknowledging our efforts in the previous revision. We have now carefully considered the reviewer's suggestions and made further improvements to enhance the clarity, consistency, and reproducibility of the manuscript.

Line 44. In order to make the Introduction suitable for a more general public, it should be mentioned that stress granules are an example of BCs.

► We have added a brief description of stress granules to make it easier to follow for a general audience:

"...Some AAs are also found to modulate the formation process of stress granules *in vivo*, a type of cytoplasmic condensates that form in response to cellular stress."

Line 59. At line 53 the authors state that they tested the effect of G on NPM1/RNA condensates, then at line 59 they mentioned other AAs ("all tested amino acids"). Although the message is clear after reading the manuscript, the reader might get lost here. I suggest rephrasing the Introduction.

► We have added the following sentence in Line 59 to improve clarity for the reader:

"In addition to glycine, we also tested other proteinogenic AAs."

Line 69. I would keep the results summarized here as general as possible. Thereby, I would avoid mentioning any K_d value, especially because the apparent dissociation constant is only introduced at line 199.

► We have removed both K_d and K_p values from the introduction to make it general to the readers.

Figure 1. If the authors want to use "dilu" and "cond" to abbreviate "dilute" and "condensed", they should define these abbreviations first.

► We have added the abbreviations "dilu" (for "dilute") and "cond" (for "condensate") to the caption of Figure 1.

Figure S5, S9. I encourage the authors to add error bars or any confidence level to all curves. Same for data points in figures S8 and S14. When possible, please also indicate the number of measurements (N) or the number of independent experiments performed.

► Measurements at individual amino acid concentrations in Figures S5, S8, and S14 were conducted in one replicate to study the binding isotherm or trends among different amino acids. We have added the information about number of measurements to the captions of relevant figures. The 95%

confidence intervals of the Langmuir-type binding model was added to Figure S5 and S9, similar to Figure 3a. The detailed fitted K_d values, along with their standard deviations, have also been provided in Figure S9.

Figure S6. It seems that the internal organization of K10-D10 condensates is heterogeneous, in the sense that some brighter sub-compartments are visible. Is this expected?

► These small, brighter regions occasionally appear in some droplets, possibly caused by small kinetically trapped aggregates. To avoid bias from these small bright spots, we calculated the average fluorescence intensity of the whole droplets.

Can the authors also include a representative image of K72-ATP condensates upon addition of 4 different AAs (figure S9) and the glycine derivatives (figure S16).

► As shown in Figures S9 and S16, we screened different amino acids and their derivatives for their effects on biomolecular condensates. Thus, we mainly used the plate reader as a high-throughput screening approach to efficiently assess whether these molecules promote or suppress condensate formation. Due to the time-consuming and labour-intensive nature of confocal microscopy experiments, we did not perform imaging for these samples. Representative images of the K72-ATP condensates at different glycine concentrations are shown in Figure S4, there is no reason to assume that the condensate appearance would be different for the other amino acids tested, given their similar K_d (Figure S9), or the absence of an effect (Figure S16).

Figure 3b,c,d. Can the authors add the data points on top of the bar plots? Are the K_d values really oscillating between 0.1 and 1, or is it just a matter of data representation (e.g., bar plots instead of box plots)?

► We have already included the individual data points in Figures 3c and 3d. For Figure 3b, we have updated Figure S9 to present the fitting curves with a 95% confidence band using the Langmuir-type binding model, similar to that in Figure 3a. The detailed fitted K_d values, along with their standard deviations, have also been provided in Figure S9. We do not find that K_d values are oscillating between 0.1 and 1. Instead, the K_d values are of the same order of magnitude (0.4 M – 1.3 M), which we also see reflected in the partition coefficients (Figure 3d).

Line 183. The authors clarified that FFsFF concentration was determined by turbidity measurements. However, the procedure is not mentioned in the Methods section.

► We have added a new section titled “Turbidity measurements” in the Methods section:

“Turbidity measurements

All turbidity measurements were performed using a plate reader (Tecan Spark M10). Absorbance was recorded across the wavelength range of 450 nm to 650 nm, with 600 nm used as the representative wavelength for turbidity. Measurements were taken immediately after transferring the entire dispersion to a 96-well UV-transparent flat-bottom plate (Nunc), right after the condensate formation by mixing all component solutions.”

Line 196. Please add a reference to justify the binding isotherm.

► We have added two references to justify the binding isotherm that we used: *Mao, Ting, et al. arXiv preprint arXiv:2404.11574 (2024)* and *Milles, Sigrid, et al. Cell 2015*.

Figure S17. I guess the authors measured the concentration of PDDA-PAA in the dilute phase in the same way as for NPM1 condensates and similar. Is PDDA or PAA fluorescently labeled? No details about the labeling or the sample preparation are mentioned in “condensate formation” (Methods).

► We have added experimental details in the Methods section describing the preparation of PDDA-PAA condensates and the measurement of PDDA concentration in the dilute phase. Briefly, the PDDA concentration in the dilute phase was determined by ¹H NMR, based on the distinct methyl group proton peak of the PDDA amine, observed between 3.1 and 3.4 ppm (*ref: Donovan, Samantha, et al. Environmental Science: Water Research & Technology 2021*):

“PDDA- PAA condensate preparation

PDDA-PAA condensates were prepared by mixing 60 mM (monomer unit) poly(diallyldimethylammonium chloride) (PDDA, 200–350 kDa) and 60 mM (monomer unit) poly(acrylic acid) sodium salt (PAA, 15 kDa) in 100 mM Tris buffer (pH 7.5), using stock solutions of 1.24 M PDDA and 3.7 M PAA in 100 mM Tris buffer (pH 7.5). After the incubation at RT for 30 mins, the condensate dispersion was centrifuged at 21130 g for 20 min at RT to spin down the condensate phase. The dilute phase was taken from the supernatant for further ¹H NMR measurements. The PDDA concentration in the dilute phase was calculated from the methyl group proton peak (3.1–3.4 ppm) in the NMR with a reference sample of a known PDDA concentration (*ref: Donovan, Samantha, et al. Environmental Science: Water Research & Technology 2021*).”

Figure S20. Figure 1 shows that NPM1-RNA condensates have a diameter ranging approximately from 1 to 5 μm (according to the reported scale bar). Although the authors claim that the RICS frames are smaller than the condensates, Figure S20 shows a RICS frame that can be bigger than an NPM1 condensate (according to the scale bar). Additionally, are guest Alexa488 molecules preferentially partitioning into the condensates? If they are homogeneously distributed in the sample, then a brightfield image is crucial to understand if the measurement has been performed in the condensate. If not, the condensates should appear brighter than the dilute phase, and, if the nanomolar concentration of Alexa488 does not allow for a clear imaging, a brightfield is again required. To clarify this, I suggest showing either the bigger frame with one or more condensates and the “zoomed” field-of-view used for RICS measurements.

► For the RICS measurements, we specifically used large condensates (diameter > 5 μm) to ensure reliable diffusion analysis within individual droplets. To obtain droplets of this size, we prepared larger sample volumes and also allowed the condensates to settle and fuse into sufficiently large droplets over time.

The Alexa 488 dye preferentially partitions into the droplets. Nevertheless, we initially used the transmitted light channel to identify suitable droplets based on size. Once appropriate droplets (> 5 μm in diameter) were identified, we zoomed in and acquired RICS frames using the fluorescence channel. A representative brightfield image and the zoomed-in field of view used for RICS measurements have been included in Figure S20.

Figure S20: (a) A representative brightfield image used to identify droplets larger than 5 μm in diameter (line patterns are artifacts in the transmitted light channel); b. After selecting an appropriate droplet, the field of view was zoomed in from the center of the droplet for further image acquisition. Shown here is an example fluorescence frame used for RICS analysis.

Line 221. The partition coefficient has been introduced at line 140. The abbreviation K_d should be reported the first time the partition coefficient is mentioned. This is also valid for other abbreviations (i.e., the one for AAs).

► The abbreviation “ K_p ” has now been introduced at line 144 when the partition coefficient was first mentioned. We have also reviewed and corrected the first occurrences of other abbreviations throughout the manuscript.

Line 372. What are A647 and A488 referring to? If A488 is referring to Alexa Fluor 488 (sometimes also mentioned as Alexa488 or Alexa 488), I encourage the authors to refer to it in a consistent way throughout the manuscript.

► A488 and A647 refer to Alexa Fluor 488 and Alexa Fluor 647, respectively. We have now used the abbreviations A488 and A647 consistently throughout the manuscript.

Line 377. Check the work “labelled”. K10 is labelled, not FITC.

► We have now corrected it.

Line 403. The authors are here describing the procedure to measure NPM1-A488 in the dilute phase, stating that the RNA is unlabelled. However, they also estimate the concentration of RNA with the same fluorescence-based analysis (line 408). They should adapt the paragraph by including both quantifications.

► We apologize for the confusion. RNA-A647 has now been removed from line 411, as RNA labelling is not required for protein NPM1 concentration measurements in the dilute phase.

Line 427. I suggest moving the description of RICS measurements to the subsequent paragraph (“Diffusion coefficient measured by RICS”). Additionally, can the authors indicate the type of fitting model they used for RICS analysis and the autocorrelation curves?

► We have moved the description of the RICS measurements to the right section. The autocorrelation curves and data analysis were performed using the PAM software (Schimpf *et al.*, *Biophysical Journal*, 2018). For quantitative analysis, we applied the 3D RICS diffusion model as described by Digman and Gratton (*Biophysical Journal*, 2009). We have added a reference to this diffusion model to the methods section.

$$G_D(\xi, \psi) = \frac{\gamma}{N} \left(1 + \frac{4D(\tau_p \xi + \tau_l \psi)}{w_0^2}\right)^{-1} \left(1 + \frac{4D(\tau_p \xi + \tau_l \psi)}{w_z^2}\right)^{-1/2}$$